# Robust Neural Rendering in the Wild with Asymmetric Dual 3D Gaussian Splatting

**Chengqi Li**[1]    **Zhihao Shi**[1]    **Yangdi Lu**[1]    **Wenbo He**[1]    **Xiangyu Xu**[2*]
[1]McMaster University    [2]Xi'an Jiaotong University

## Abstract

3D reconstruction from in-the-wild images remains a challenging task due to inconsistent lighting conditions and transient distractors. Existing methods typically rely on heuristic strategies to handle the low-quality training data, which often struggle to produce stable and consistent reconstructions, frequently resulting in visual artifacts. In this work, we propose **AsymGS**, a novel framework that leverages the stochastic nature of these artifacts: they tend to vary across different training runs due to minor randomness. Specifically, our method trains two 3D Gaussian Splatting (3DGS) models in parallel, enforcing a consistency constraint that encourages convergence on reliable scene geometry while suppressing inconsistent artifacts. To prevent the two models from collapsing into similar failure modes due to confirmation bias, we introduce a divergent masking strategy that applies two complementary masks: a multi-cue adaptive mask and a self-supervised soft mask, which leads to an asymmetric training process of the two models, reducing shared error modes. In addition, to improve the efficiency of model training, we introduce a lightweight variant called Dynamic EMA Proxy, which replaces one of the two models with a dynamically updated Exponential Moving Average (EMA) proxy, and employs an alternating masking strategy to preserve divergence. Extensive experiments on challenging real-world datasets demonstrate that our method consistently outperforms existing approaches while achieving high efficiency. See the project website at `https://steveli88.github.io/AsymGS`.

## 1   Introduction

3D scene reconstruction from multiple views is a fundamental problem in computer vision. Recent advances such as Neural Radiance Fields (NeRF) [15] and 3D Gaussian Splatting (3DGS) [8] have achieved impressive rendering quality by learning volumetric or point-based scene representations from posed images. However, these methods typically assume that training images exhibit consistent illumination and minimal occlusion, which are rarely satisfied in real-world settings.

In-the-wild images are often captured under varying lighting conditions and contain transient distractors such as pedestrians or vehicles; these factors introduce substantial noise into the supervision signal, leading to degraded reconstruction quality and visual artifacts. While several recent works have attempted to address these challenges [14, 19, 18, 26, 10, 20, 2, 13], they largely rely on heuristic strategies to suppress the effects of corrupted supervision from low-quality training data, such as per-image appearance embeddings that are only weakly or indirectly supervised through photometric losses [14, 26], or hand-crafted rules to filter outlier training signals [19]. As a result, such approaches often lack stability and generalizability, which leads to artifact-prone reconstructions.

To bridge this gap, we propose a new framework, called **AsymGS**, which is motivated by a key empirical observation: *artifacts arising from low-quality in-the-wild training data are typically*

---

*Corresponding author

39th Conference on Neural Information Processing Systems (NeurIPS 2025).

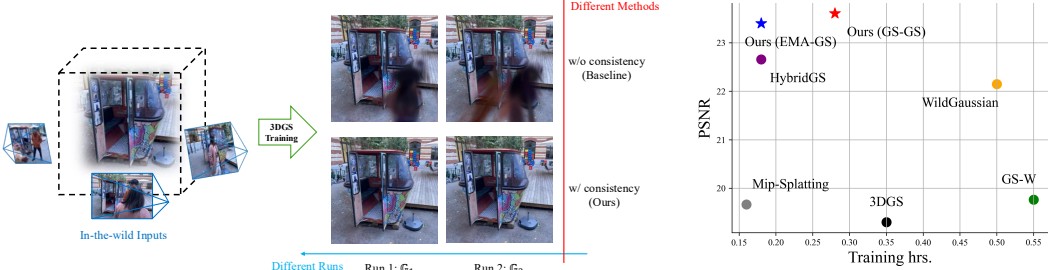

Figure 1: **Left:** The key insight of this work is that artifacts arising from low-quality in-the-wild inputs are typically stochastic across different runs of the same model (Baseline: Run 1 *vs.* Run 2). This motivates the design of the Asymmetric Dual 3DGS framework, which enhances true scene structure while suppressing errors through cross-model consistency (w/ consistency). **Right:** Our method compares favorably against the state-of-the-art approaches in terms of reconstruction quality while maintaining high training efficiency. Results are on the NeRF On-the-go dataset [18].

*stochastic*. In other words, the artifacts vary randomly across different runs of the same model with only minor training perturbations, such as data order shuffling (see Figure 1-Left). This suggests that enforcing consistency between independently trained 3DGS can help suppress unreliable or spurious signals in the in-the-wild training images. To this end, we introduce a dual-model architecture where two 3DGS models are trained concurrently with a consistency constraint, following the intuition that true scene structure should be consistently reconstructed across different model runs, while the artifacts induced by low-quality data tend to diverge.

Nevertheless, naively training two models in parallel can lead to confirmation bias, where both models reinforce the same errors. To encourage more divergent error modes and mitigate confirmation bias, we introduce a divergent masking strategy: applying distinct masks to each model that emphasize complementary factors for filtering out transient or distracting content. One mask is learned in a self-supervised manner based on feature-level similarity between predicted and ground truth images, while the other, called multi-cue adaptive mask, uses stereo-based correspondence to identify likely distracting regions. These complementary filtering schemes encourage the two 3DGS models to focus on different static aspects of the scene. Consequently, this asymmetric strategy leads to divergent and complementary optimization paths and reduces shared error modes. The final reconstruction is then guided by the agreement between the two models, which reliably captures consistent and accurate scene structures while suppressing artifacts.

While the dual-model framework effectively improves the reconstruction quality, it introduces notable computational overhead in the training process. To mitigate this, we introduce a lightweight variant, called Dynamic EMA Proxy, which replaces the second 3DGS model with a dynamic, training-free Exponential Moving Average (EMA) copy of the primary model. Unlike standard EMA [4], our Dynamic EMA proxy is specifically designed to track the evolving nature of 3DGS representations, accounting for Gaussian densification and pruning. Since only one model is actively trained in this setup, which no longer allows independent masks for two models, we additionally design an alternating masking strategy that alternates between the two masks, maintaining divergent training signals and mitigating confirmation bias.

Our contributions are as follows: 1) We propose a AsymGS framework for in-the-wild 3D scene reconstruction. By enforcing consistency constraints between two 3DGS models with complementary masks, our framework significantly improves the robustness and accuracy of scene representations. 2) We develop a divergent masking strategy by introducing different masking mechanisms for each 3DGS model, which handle various types of distractors and promote divergent optimization paths to mitigate confirmation bias. 3) To address the computational overhead of the dual-model framework, we introduce a Dynamic EMA Proxy, coupled with an alternating masking strategy, which effectively improves training efficiency. 4) We conduct extensive evaluations across a diverse set of in-the-wild 3D scene reconstruction datasets, demonstrating that our method consistently achieves state-of-the-art performance and efficiency, highlighting its robustness and generality.

## 2   Related work

**3D Scene Reconstruction.** Neural Radiance Fields (NeRF) [15] revolutionizes photorealistic novel view synthesis by modeling scenes as continuous functions that map 3D coordinates to color and density. More recently, 3D Gaussian Splatting (3DGS) [8, 25] has gained attention as a real-time

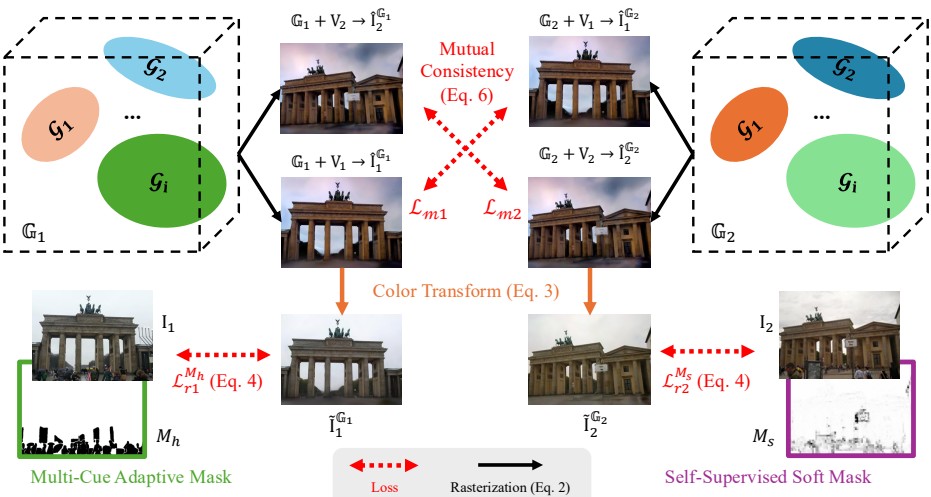

Figure 2: Overview of the AsymGS framework. Two 3DGS models $\mathbb{G}_1$ and $\mathbb{G}_2$ are concurrently optimized with the reconstruction loss $\mathcal{L}_{r1}^{\mathbf{M}_h}$ and $\mathcal{L}_{r2}^{\mathbf{M}_s}$ (Eq. 4), along with the mutual consistency loss $\mathcal{L}_{m1}$ and $\mathcal{L}_{m2}$ (Eq. 6). In addition, we apply a mask loss (Eq. 7) for learning soft mask in a self-supervised manner. For improved efficiency, we also propose an EMA version of our framework by replacing $\mathbb{G}_2$ with a dynamic EMA proxy. Both the mask loss and the EMA proxy have been omitted here for clarity. Note that the color transform in this figure is for illustration purpose, which undergoes a rasterization process in practice as introduced in Section 3.1.

alternative, representing scenes with optimizable Gaussian primitives. While effective, both methods assume static scenes to enforce multi-view consistency, an assumption often violated in in-the-wild settings due to varying illumination and transient objects, limiting their practical applicability.

**3D Scene Reconstruction in the Wild.** NeRF-W [14] first addressed 3D scene reconstruction in the wild with a modular architecture combining a learnable appearance embedding and an uncertainty map to suppress distractors—an approach that has since become standard. NeRF On-the-go [18] builds on this by using DINOv2 feature [16] residuals to construct uncertainty maps. Robust-NeRF [19] tackles noisy training images through a robust loss function. GS-W [26], Wild-GS [24] and WildGaussian [11] extend learnable embeddings with both global and per-Gaussian local embeddings for fine-grained appearance modeling. SpotlessSplats [20] introduces a learnable mask based on thresholded and dilated residuals, while SWAG [2] adds a view-dependent opacity term per Gaussian to identify transient distractors. HybridGS [13] employs a dual-model setup (3DGS for static content and 2DGS [5] for dynamic distractors) to learn an accurate uncertainty map iteratively. Despite their contributions, these methods largely rely on heuristic strategies to suppress corrupted supervision signals from low-quality training data. For instance, the per-image appearance embeddings are only weakly or indirectly supervised through photometric losses [14, 26], and outlier filtering is often governed by hand-crafted rules [19]. As a result, these approaches frequently suffer from instability and produce reconstructions with noticeable artifacts. In this work, we propose a principled framework, AsymGS, to reduce such artifacts and improve reconstruction stability, based on the key observation that many artifacts exhibit stochastic behavior and are not consistent across different training runs. Our method explicitly exploits this property to enhance robustness and achieve high-fidelity reconstructions.

# 3 Method

An overview of the proposed algorithm is shown in Figure 2. Please refer to the caption for details.

## 3.1 Preliminaries

3D Gaussian Splatting (3DGS) [8] represents a 3D scene as a set of $N$ 3D anisotropic Gaussians $\mathbb{G} = \{\mathcal{G}_i\}_{i=1}^{N}$. Each Gaussian $\mathcal{G}_i$ is parameterized by a centroid $\mathbf{X}_i$, a covariance matrix $\mathbf{\Sigma}_i$, an opacity $\alpha_i$, and a set of spherical harmonic (SH) coefficients $\theta_i$ for view-dependent color representation. For

rendering with a viewing camera $\mathbf{V}_j$, we project both centroids of Gaussians and the covariance matrix onto the 2D image plane as $\mathbf{x}_{ij}$ and $\boldsymbol{\Sigma}_{ij}$, respectively. The projected opacity $\alpha_{ij}$, which is a function of 2D image plane coordinate $\mathbf{y}$, can then be defined as below:

$$\alpha_{ij}(\mathbf{y}) = \alpha_i \cdot \exp\left[-\frac{1}{2}(\mathbf{y} - \mathbf{x}_{ij})^T {\boldsymbol{\Sigma}_{ij}}^{-1}(\mathbf{y} - \mathbf{x}_{ij})\right] \tag{1}$$

The color $\mathbf{C}$ of a pixel located at $\mathbf{y}$ for camera $\mathbf{V}_j$ is computed using the $\alpha$-blending with the following formula:

$$\mathbf{C}(\mathbf{y}, \mathbf{V}_j, \mathbb{G}) = \sum_{i=1}^{N} \mathbf{c}_{ij}\alpha_{ij}(\mathbf{y}) \prod_{k=1}^{i-1}(1 - \alpha_{kj}(\mathbf{y})), \quad \mathbf{c}_{ij} = \mathrm{SH}\left(\mathbf{r}_{ij}, \theta_i\right) \tag{2}$$

where $\mathbf{r}_{ij}$ is the ray direction from the $j$-th camera center to the $i$-th Gaussian centroid, and $\mathbf{c}_{ij}$ is the corresponding color of the observed Gaussian primitive obtained using spherical harmonic (SH) function. Performing Eq. 2 for every pixel on the image plane constitutes the rasterization process, which results in the rendered image $\hat{\mathbf{I}}_j^{\mathbb{G}} = \mathrm{Rasterize}(\mathbb{G}, \mathbf{V}_j)$.

**View-dependent appearance modeling.** To address appearance variations in in-the-wild data, we follow the approach of WildGaussian [11] to adaptively adjust the observed color of the Gaussian primitives to account for the view-dependent factors, such as the varying illumination across images captured at different times of day.

This adjustment is conditioned on both the per-Gaussian appearance embedding $\mathbf{p}_i$ and the per-view appearance embedding $\mathbf{q}_j$. Specifically, $\mathbf{p}_i$, $\mathbf{q}_j$, and $\mathbf{c}_{ij}$ are sent into an MLP $f$ to predict affine transformation parameters:

$$(a, b) = f(\mathbf{p}_i, \mathbf{q}_j, \mathbf{c}_{ij}), \quad \tilde{\mathbf{c}}_{ij} = a \cdot \mathbf{c}_{ij} + b, \tag{3}$$

where $a$ and $b$ are three-dimensional outputs corresponding to RGB channels. The transformed color $\tilde{\mathbf{c}}_{ij}$ is then used to replace $\mathbf{c}_{ij}$ in the blending process described by Eq. 2, and the resulting view-dependent image is denoted as $\tilde{\mathbf{I}}_j^{\mathbb{G}} = \mathrm{Rasterize}_{\mathrm{dep}}(\mathbb{G}, \mathbf{V}_j)$. During training, $\mathbf{p}_i$, $\mathbf{q}_j$, $f$, and 3DGS parameters are jointly optimized.

## 3.2 Dual 3DGS

A central insight of this work is that artifacts arising from in-the-wild training data are typically stochastic in nature. When two 3DGS models are trained on the same scene but with different view sampling orders, their static scene representations remain consistent, whereas their renderings could diverge in regions affected by outliers. An example is shown in Figure 1-Left.

Motivated by this observation, we introduce a framework with two 3DGS, where each model is trained with a different sampling order, and a consistency constraint is enforced between their renderings. Specifically, we maintain two sets of Gaussians, $\mathbb{G}_1$ and $\mathbb{G}_2$, to represent the same scene. In each training iteration, we independently sample two views from separate training view lists, yielding two viewing cameras $\mathbf{V}_1$ and $\mathbf{V}_2$, along with their corresponding ground-truth images, $\mathbf{I}_1$ and $\mathbf{I}_2$.

Similar to 3DGS [8], we train $\mathbb{G}_1$ and $\mathbb{G}_2$ with reconstruction objectives defined as:

$$\mathcal{L}_{r1}^{\mathbf{M}} = \mathcal{L}_{\mathrm{recon}}(\tilde{\mathbf{I}}_1^{\mathbb{G}_1}, \mathbf{I}_1, \mathbf{M}), \quad \mathcal{L}_{r2}^{\mathbf{M}} = \mathcal{L}_{\mathrm{recon}}(\tilde{\mathbf{I}}_2^{\mathbb{G}_2}, \mathbf{I}_2, \mathbf{M}), \tag{4}$$

$$\mathcal{L}_{\mathrm{recon}}(\tilde{\mathbf{I}}, \mathbf{I}, \mathbf{M}) = \lambda \cdot \mathrm{DSSIM}(\mathbf{M} \odot \tilde{\mathbf{I}}, \mathbf{M} \odot \mathbf{I}) + (1 - \lambda) \cdot \|\mathbf{M} \odot \tilde{\mathbf{I}} - \mathbf{M} \odot \mathbf{I}\|_1, \tag{5}$$

where $\tilde{\mathbf{I}}_j^{\mathbb{G}_n} = \mathrm{Rasterize}_{\mathrm{dep}}(\mathbb{G}_n, \mathbf{V}_j)$ is the rendered image for the $n$-th 3DGS model from viewpoint $\mathbf{V}_j$ using Eq. 2 and 3. $\mathbf{M}$ is a spatial mask to filter out transient distracting regions, such as pedestrians or moving vehicles, which will be detailed in Section 3.3. $\odot$ denotes element-wise multiplication. DSSIM represents the structural dissimilarity index measure [23]. $\lambda$ is a hyperparameter to balance the DSSIM and the $L_1$ terms.

**Mutual consistency.** Since $\mathbb{G}_1$ and $\mathbb{G}_2$ represent the same underlying scene, their renderings from the same camera viewpoint should stay close. This motivates us to define a mutual consistency regularization as:

$$\mathcal{L}_{m1} = \|\hat{\mathbf{I}}_1^{\mathbb{G}_2} - \hat{\mathbf{I}}_1^{\mathbb{G}_1}\|_1, \quad \mathcal{L}_{m2} = \|\hat{\mathbf{I}}_2^{\mathbb{G}_1} - \hat{\mathbf{I}}_2^{\mathbb{G}_2}\|_1 \tag{6}$$

where $\hat{\mathbf{I}}_j^{\mathbb{G}_n} = \text{Rasterize}(\mathbb{G}_n, \mathbf{V}_j)$ is the view-dependent rendering obtained via Eq. 2. We emphasize that this consistency constraint is performed over $\hat{\mathbf{I}}_j^{\mathbb{G}_n}$ instead of $\tilde{\mathbf{I}}_j^{\mathbb{G}_n}$, because $\hat{\mathbf{I}}_j^{\mathbb{G}_n}$ captures the intrinsic appearance of the 3D scene, whereas $\tilde{\mathbf{I}}_j^{\mathbb{G}_n}$ is affected by dynamic lighting. This strategy provides a principled way to preserve static structures while suppressing spurious signals, which enables more robust and reliable reconstruction.

Note that we only use the $L_1$ loss for consistency regularization in Eq. 6 as incorporating the DSSIM loss adversely affects performance in our experiments. Furthermore, we empirically find that incorporating consistency regularization too early in training can hinder convergence, as both models may still be dominated by noise and unstable geometry. To address this, we adopt a progressive strategy: we first allow the two models to be trained independently for a number of warm-up iterations, during which they develop their own estimates of the static scene. Once their reconstructions become sufficiently stable, we introduce the consistency loss to encourage convergence on shared, reliable structures.

### 3.3 Asymmetric Dual 3DGS

While the above framework offers improved consistency through mutual supervision, its symmetric design, where both 3DGS models are trained in the same manner using the reconstruction loss in Eq. 5, poses a risk of confirmation bias: both models may converge toward the same reconstruction errors due to their similar optimization signals.

To address this issue, we propose an Asymmetric Dual 3DGS variant, where each model is trained with a distinct masking strategy that emphasizes complementary criteria for filtering out transient or distracting content. This encourages divergent error patterns, enhances robustness, and mitigates confirmation bias. Specifically, we use a Multi-Cue Adaptive Mask and a Self-Supervised Soft Mask.

**Multi-Cue Adaptive Mask** ($\mathbf{M}_h$). As illustrated in Figure 3, $\mathbf{M}_h$ is a hard binary mask (1 indicates static regions and 0 indicates distractors) that identifies transient and distracting regions by integrating multiple cues, including semantic segmentation, stereo correspondence, pixel-level residuals, and feature-level residuals.

We begin by applying the Segment Anything (SAM) model [9, 12] to partition each image into semantically coherent regions. To detect static content, we perform multi-view stereo matching across the training images with COLMAP [21]. Semantic regions are considered static if they contain a sufficient number of valid correspondences (we empirically choose a threshold of 3 matches). Among the remaining regions, we identify transient distractors by analyzing reconstruction residuals. For each region, we compute pixel-level residuals, *i.e.*, the $L_1$ error between the rendered and ground-truth images, and feature-level residuals, *i.e.*, the cosine distance between DINOv2-encoded feature maps [16] of the rendered and ground-truth images. Regions with above-average residuals in both metrics are classified as distractors and masked out during training. This multi-cue approach offers higher robustness than prior methods that rely on single cues [19, 18, 14, 17], which generalizes more effectively across diverse in-the-wild scenes. See Algorithm 1 in the supplementary material for full details of the Multi-Cue Adaptive Mask.

**Self-Supervised Soft Mask** ($\mathbf{M}_s$). To complement the rule-based hard mask $\mathbf{M}_h$, we introduce a learnable soft mask $\mathbf{M}_s$, whose values range between 0 and 1. Unlike the static $\mathbf{M}_h$, this soft mask is optimized jointly with the model and adapts throughout training. The objective for $\mathbf{M}_s$ is derived from the cosine similarity between DINOv2 feature maps of the ground-truth image $\mathbf{F}$ and the rendered image $\tilde{\mathbf{F}}$:

$$\mathcal{L}_{\text{mask}} = \|\mathbf{M}_s - f_{\text{interp}}(\text{CosineSimilarity}(\mathbf{F}, \tilde{\mathbf{F}}))\|_1, \tag{7}$$

where $f_{\text{interp}}$ denotes spatial interpolation to match the training image resolution. This formulation is self-supervised, requiring no ground-truth masks. We initialize $\mathbf{M}_s$ as an all-one tensor, allowing the model to gradually refine the mask as training progresses (Figure 3).

As shown in Figure 3, the hard mask $\mathbf{M}_h$ is more definitive with clearer boundaries but can be over-confident, potentially missing certain transient objects. In contrast, the soft mask $\mathbf{M}_s$ is often more sensitive to subtle distractors and better captures ambiguous regions, thus providing complementary information to $\mathbf{M}_h$.

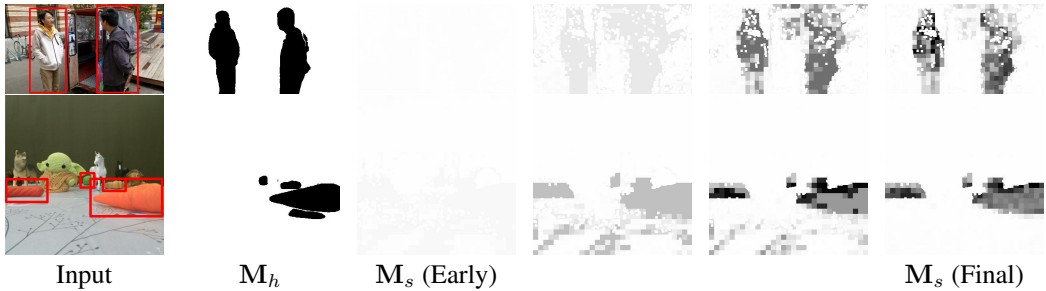

| Input | $\mathbf{M}_h$ | $\mathbf{M}_s$ (Early) | | | $\mathbf{M}_s$ (Final) |

Figure 3: Comparisons of hard and soft masks. Distractors are highlighted in red boxes in the input. The right four columns show the evolving of the self-supervised soft mask across different epochs.

By combining the loss terms in Eq. 5, 6, 7, the overall objective for our AsymGS can be written as:

$$\mathcal{L} = \mathcal{L}_{r1}^{\mathbf{M}_h} + \mathcal{L}_{r2}^{\mathbf{M}_s} + \lambda_m(\mathcal{L}_{m1} + \mathcal{L}_{m2}) + \lambda_{\text{mask}}\mathcal{L}_{\text{mask}}, \qquad (8)$$

where $\lambda_m$ and $\lambda_{\text{mask}}$ are hyperparameters for balancing loss terms. Since $\mathbf{M}_s$ dynamically evolves during training and intrinsically differs from the fixed $\mathbf{M}_h$, training one model with $\mathcal{L}_{r1}^{\mathbf{M}_h}$ and the other with $\mathcal{L}_{r2}^{\mathbf{M}_s}$ in our AsymGS framework introduces complementary inductive biases. This asymmetry promotes diverse learning dynamics, making it less likely for the two models to converge on the same reconstruction error, thereby reducing confirmation bias and enhancing overall robustness.

### 3.4 Dynamic EMA proxy

While the AsymGS framework significantly improves robustness and reconstruction quality, it requires simultaneous training of two 3DGS models, which introduces considerable computational overhead and undermines the fast training advantage of 3DGS. To address this issue, we propose a more efficient alternative by replacing one of the two models with a dynamic EMA proxy. Moreover, an alternating masking strategy is introduced to counteract confirmation bias. This design retains the benefits of dual-model regularization while significantly reducing computation cost.

Let $\mathbb{G}_1$ denote the set of Gaussians actively optimized during training, and $\mathbb{G}_{\text{EMA}}$ its EMA counterpart, updated at each training step by:

$$\mathbb{G}_{\text{EMA}}^t = \beta \cdot \mathbb{G}_{\text{EMA}}^{t-1} + (1 - \beta) \cdot \mathbb{G}_1^t, \quad \mathbb{G}_{\text{EMA}}^0 = \mathbb{G}_1^0, \qquad (9)$$

where $t$ and $t-1$ denote the current and previous training iterations, respectively. Here, we slightly abuse set notation for simplicity: the weighted summation between $\mathbb{G}_1$ and $\mathbb{G}_{\text{EMA}}$ is performed element-wise over corresponding Gaussian attributes, such as the centroids, opacities, and SH coefficients. We then rewrite the consistency regularization in Eq. 6 with the EMA proxy as follows:

$$\mathcal{L}_{me} = \|\hat{\mathbf{I}}_1^{\mathbb{G}_{\text{EMA}}} - \hat{\mathbf{I}}_1^{\mathbb{G}_1}\|_1, \qquad (10)$$

where $\hat{\mathbf{I}}_1^{\mathbb{G}_{\text{EMA}}} = \text{Rasterize}(\mathbb{G}_{\text{EMA}}, \mathbf{V}_1)$ is the rendering of the EMA Gaussians from view $\mathbf{V}_1$. Since only one 3DGS model requires gradient updates, and the EMA update is a simple weighted average, this approach greatly improves training efficiency while preserving the benefits of dual-model consistency.

**Dynamic update.** Standard EMA is primarily designed for neural networks, where the number of parameters is typically fixed throughout training [22, 4]. However, applying it to 3DGS presents unique challenges, as the number of Gaussians dynamically changes during training due to operations such as cloning, splitting, and pruning [8].

To support this dynamic data structure, we develop a dynamic EMA mechanism by introducing the following rules: 1) **Cloning**: When a Gaussian is cloned, its EMA attributes are also cloned. 2) **Pruning**: When a Gaussian is pruned, its EMA counterpart is removed as well. 3) **Splitting**: When a Gaussian splits into two, attributes that undergo discontinuous changes, *i.e.*, the centroids and variances, are reinitialized in the EMA according to the values of the split Gaussians. The remaining attributes (e.g., opacities and SH coefficients) are directly inherited from the original EMA representation.

**Alternating masking strategy.** Since only one model is trainable in our EMA framework, the original asymmetric training strategy used in Eq. 8 (*i.e.*, $\mathbf{M}_h$ and $\mathbf{M}_s$) is not directly applicable.

Table 1: Quantitative results on the NeRF On-the-go dataset [18]. Efficiency is reported in terms of average training hours per scene. The best and second-best results are highlighted in **bold** and underline, respectively.

| Scene | High Occlusion | | | Medium Occlusion | | | Low Occlusion | | | |
|---|---|---|---|---|---|---|---|---|---|---|
| Method | PSNR↑ | SSIM↑ | LPIPS↓ | PSNR↑ | SSIM↑ | LPIPS↓ | PSNR↑ | SSIM↑ | LPIPS↓ | Hrs. |
| RobustNeRF [19] | 20.60 | 0.602 | 0.379 | 21.72 | 0.741 | 0.248 | 16.60 | 0.407 | 0.480 | - |
| NeRF On-the-go [18] | 22.37 | 0.753 | 0.212 | 22.50 | 0.780 | 0.205 | 20.13 | 0.627 | 0.287 | 43 |
| 3DGS [8] | 19.03 | 0.649 | 0.340 | 19.19 | 0.709 | 0.220 | 19.68 | 0.649 | 0.199 | 0.35 |
| Mip-Splatting [25] | 19.25 | 0.664 | 0.333 | 19.73 | 0.684 | 0.279 | 20.03 | 0.661 | 0.195 | 0.16 |
| GS-W [26] | 18.52 | 0.645 | 0.335 | 21.04 | 0.737 | 0.208 | 19.75 | 0.660 | 0.287 | 0.55 |
| WildGaussian [11] | 23.03 | 0.771 | 0.172 | 22.80 | 0.811 | 0.092 | 20.62 | 0.658 | 0.235 | 0.50 |
| SLS-mlp [20] | 21.92 | 0.710 | 0.222 | 22.79 | 0.817 | 0.162 | 20.02 | 0.596 | 0.276 | - |
| HybridGS [13] | 23.05 | 0.768 | 0.204 | 23.51 | 0.830 | 0.160 | 21.42 | 0.684 | 0.268 | 0.18 |
| Ours (GS-GS) | **24.34** | **0.825** | **0.150** | **24.56** | **0.872** | **0.090** | **21.91** | **0.728** | 0.189 | 0.28 |
| Ours (EMA-GS) | 24.12 | 0.818 | 0.154 | 24.32 | 0.864 | 0.090 | 21.77 | 0.722 | **0.162** | 0.18 |

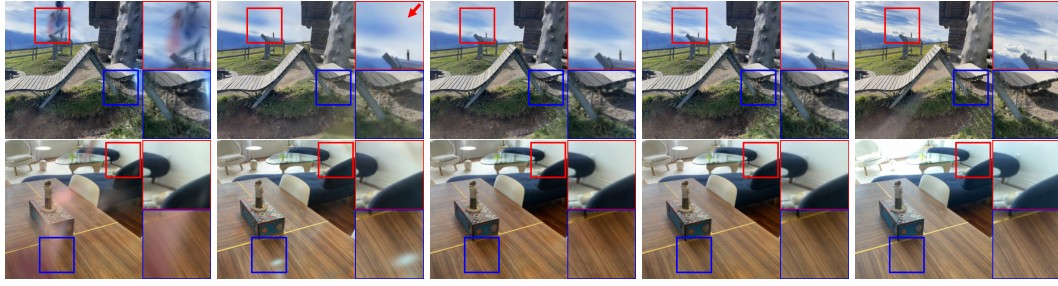

| Mip-Splatting [25] | HybridGS [13] | Our (EMA-GS) | Ours (GS-GS) | Ground Truth |

Figure 4: Qualitative results on the NeRF On-the-go [18] (top) and the RobustNeRF [19] (bottom) datasets.

Instead, we propose an alternating masking strategy by switching between the hard mask $\mathbf{M}_h$ and the soft mask $\mathbf{M}_s$ for training $\mathbb{G}_1$, which retains the complementary advantages from both decisive, rule-based filtering and adaptive, learned filtering. The final loss for our dynamic EMA framework can be written as:

$$\mathcal{L} = \mathcal{L}_{r1}^{\mathbf{M}_{h/s}} + \lambda_m \mathcal{L}_{me} + \lambda_{\text{mask}} \mathcal{L}_{\text{mask}}, \tag{11}$$

where $\mathbf{M}_{h/s}$ indicates alternating between masks. This strategy essentially injects randomness into the EMA update process, promoting diversity in optimization and reducing overfitting to erroneous supervision signals. Note that we also explored other forms of randomness, including randomly mixing up EMA renderings with ground truth and applying random dropout of Gaussian primitives. Nevertheless, we empirically find that alternating masking remains the most effective approach.

**Discussion.** Our approach is related to prior works that also leverage EMA, such as [22] and [4], which apply EMA to neural networks for tasks like semi-supervised or unsupervised image classification. However, our method diverges in key aspects: unlike these methods that operate in the context of neural networks, we apply EMA to 3DGS, a dynamic representation where the number of Gaussians evolves throughout training. This necessitates our proposed dynamic EMA mechanism, which adapts EMA updates to structural changes in the learned scene representation. Additionally, we introduce an alternating masking strategy to preserve the benefits of asymmetric training even with a single learnable model. These innovations mark significant departures from conventional EMA usage and highlight the contributions of this work.

## 4 Experiments

We evaluate our method on three in-the-wild datasets: the NeRF On-the-go dataset [18], the Robust-NeRF dataset [19], and the PhotoTourism dataset [6]. NeRF On-the-go and RobustNeRF mainly suffer from transient distractors. PhotoTourism contains both distractors and dynamic lighting. We denote our full dual-model approach as GS-GS, and the efficient variant with the EMA proxy as EMA-GS. Implementation details can be found in Section E of the supplementary material.

Table 2: Quantitative results on the RobustNeRF dataset [19]. Efficiency is reported in terms of average training hours per scene. The best and second-best results are highlighted in **bold** and underline, respectively.

| Scene | Statue | | | Android | | | Yoda | | | Crab | | | |
|---|---|---|---|---|---|---|---|---|---|---|---|---|---|
| Method | PSNR↑ | SSIM↑ | LPIPS↓ | PSNR↑ | SSIM↑ | LPIPS↓ | PSNR↑ | SSIM↑ | LPIPS↓ | PSNR↑ | SSIM↑ | LPIPS↓ | Hrs. |
| RobustNeRF [19] | 20.60 | 0.760 | 0.150 | 23.28 | 0.750 | 0.130 | 29.78 | 0.820 | 0.150 | - | - | - | - |
| NeRF On-the-go [18] | 21.58 | 0.770 | 0.240 | 23.50 | 0.750 | 0.210 | 29.96 | 0.830 | 0.240 | - | - | - | - |
| 3DGS [8] | 21.02 | 0.810 | 0.160 | 23.11 | 0.810 | 0.130 | 26.33 | 0.910 | 0.140 | 29.74 | - | - | - |
| Mip-Splatting [25] | 22.08 | 0.860 | 0.135 | 23.45 | 0.801 | 0.106 | 27.96 | 0.933 | 0.136 | 29.18 | 0.929 | 0.129 | 0.14 |
| GS-W [25] | 21.99 | 0.862 | 0.102 | 24.23 | 0.824 | 0.090 | 32.74 | 0.957 | 0.084 | 33.22 | 0.952 | 0.088 | 0.37 |
| WildGaussian [11] | 23.25 | 0.886 | 0.105 | 24.57 | 0.827 | 0.085 | 32.84 | 0.956 | 0.091 | 32.81 | 0.952 | 0.092 | 0.82 |
| SLS-mlp [20] | 22.54 | 0.840 | 0.130 | 25.05 | 0.850 | 0.090 | 33.66 | 0.960 | 0.100 | 34.43 | - | - | - |
| HybridGS [13] | 22.93 | 0.870 | 0.100 | 25.15 | 0.850 | 0.070 | 35.32 | 0.960 | **0.070** | 35.17 | 0.960 | 0.080 | - |
| Ours (GS-GS) | 23.47 | **0.894** | 0.097 | **25.61** | **0.857** | 0.071 | **37.18** | **0.969** | 0.074 | **36.18** | **0.964** | **0.078** | 0.31 |
| Ours (EMA-GS) | **23.49** | 0.890 | **0.096** | 25.47 | 0.849 | **0.068** | 36.50 | 0.967 | 0.077 | 35.60 | 0.961 | 0.080 | 0.21 |

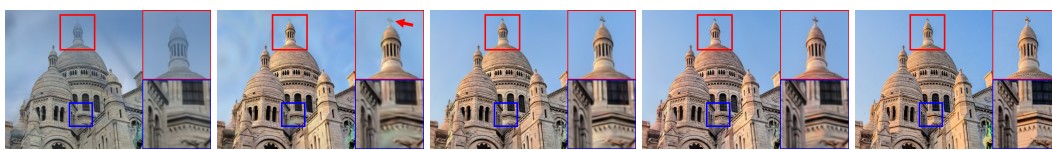

Mip-Splatting [25]   WildGaussian [11]   Our (EMA-GS)   Ours (GS-GS)   Ground Truth

Figure 5: Qualitative results on the PhotoTourism dataset [6].

## 4.1 Comparison with SOTA

As shown in Tables 1-3, the proposed AsymGS framework, in both the GS-GS and EMA-GS setups, consistently outperforms all baselines by a significant margin across all evaluation datasets, highlighting the robustness and generality of our approach. Visual comparisons are provided in Figure 4.

In terms of training efficiency, our GS-GS model trains in an average of 30 minutes per scene on the NeRF On-the-go and RobustNeRF datasets, while the EMA proxy further reduces training time by one-third. For the PhotoTourism dataset, which features high-resolution imagery and dynamic lighting, EMA-GS cuts training time from 7.2 hours to 2.9 hours compared to the previous SOTA [11] while achieving better reconstruction quality.

## 4.2 Ablation study

**Dual 3DGS framework.** As shown in Table 4, the dual 3DGS framework, whether implemented directly or via an EMA proxy, consistently outperforms the single-model baseline across three datasets (L1 *vs.* L5 & L12 in Table 4). Importantly, the mutual consistency loss is a crucial component. Removing it reduces the framework to a simple ensemble, which is shown to be ineffective, leading to an average drop of 0.5 dB in PSNR and consistent degradation in SSIM and LPIPS (L5 *vs.* L9, L12 *vs.* L16).

Additionally, the GS-GS setup achieves better performance than its EMA proxy counterpart (L5 *vs.* L12). We attribute this to two main factors: first, the GS-GS setup enables both models to be actively updated, effectively doubling the training iterations; second, the EMA proxy introduces confirmation bias due to its model accumulation nature, which limits its ability to correct erroneous predictions.

**Distractor modeling.** The results in Table 4 further show that applying masks significantly improves reconstruction performance especially when distractors occupy a large portion of the input image. (L5 *vs.* L6, L12 *vs.* L13).

Either $\mathbf{M}_h$ or $\mathbf{M}_s$ independently improves performance over the base model (for $\mathbf{M}_h$, L2 *vs.* L3, L6 *vs.* L7, L13 *vs.* L14; for $\mathbf{M}_s$, L2 *vs.* L4, L6 *vs.* L8, L13 *vs.* L15), indicating that both masks are effective, although they have different characteristics. This observation is further supported by the visual results in Figure 3, where the hard-selected mask performs well in simpler scenes with clearly defined regions (Figure 3-Top), while the self-supervised mask excels in more complex scenes containing multiple or ambiguous distractors (Figure 3-Bottom). Moreover, combining both masks leads to additional gains, confirming their complementary roles (L5 *vs.* L7 & L8, L12 *vs.* L14 & L15).

Table 3: Quantitative results on the PhotoTourism dataset [6]. Efficiency is reported in terms of average training hours per scene. The best and second-best results are highlighted in **bold** and underline, respectively. We did not compare with HybridGS [13] on PhotoTourism, as it does not consider varying illumination, and cannot be customized or retrained due the absence of training code.

| Scene | Brandenburg Gate | | | Sacre Coeur | | | Trevi Fountain | | | |
|---|---|---|---|---|---|---|---|---|---|---|
| Method | PSNR↑ | SSIM↑ | LPIPS↓ | PSNR↑ | SSIM↑ | LPIPS↓ | PSNR↑ | SSIM↑ | LPIPS↓ | Hrs. |
| NeRF [15] | 18.90 | 0.882 | 0.138 | 15.60 | 0.846 | 0.163 | 16.14 | 0.696 | 0.282 | - |
| 3DGS [8] | 19.37 | 0.880 | 0.141 | 17.44 | 0.835 | 0.204 | 17.58 | 0.709 | 0.266 | 2.2 |
| Mip-Splatting [25] | 20.01 | 0.877 | 0.166 | 17.54 | 0.831 | 0.203 | 17.36 | 0.684 | 0.319 | 2.3 |
| NeRF-W [14] | 24.17 | 0.891 | 0.152 | 19.20 | 0.803 | 0.192 | 18.97 | 0.698 | 0.265 | 164 |
| Ha-NeRF [1] | 24.04 | 0.887 | 0.139 | 20.02 | 0.801 | 0.171 | 20.18 | 0.691 | 0.223 | 452 |
| K-Planes [3] | 25.49 | 0.879 | 0.224 | 20.61 | 0.774 | 0.265 | 22.67 | 0.714 | 0.317 | 0.6 |
| RefinedFields [7] | 26.64 | 0.886 | - | 22.26 | 0.817 | - | 23.42 | 0.737 | - | 150 |
| GS-W [26] | 23.51 | 0.897 | 0.166 | 19.39 | 0.825 | 0.211 | 20.06 | 0.723 | 0.274 | 1.2 |
| SWAG [2] | 26.33 | 0.929 | 0.139 | 21.16 | 0.860 | 0.185 | 23.10 | **0.815** | 0.208 | 0.8 |
| WildGaussian [11] | 27.77 | 0.927 | 0.133 | 22.56 | 0.859 | 0.177 | 23.63 | 0.766 | 0.228 | 7.2 |
| Ours (GS-GS) | **28.56** | **0.938** | **0.109** | **23.78** | **0.887** | **0.139** | **24.52** | 0.790 | **0.202** | 5.3 |
| Ours (EMA-GS) | 28.50 | 0.937 | 0.115 | 23.37 | 0.882 | 0.150 | 23.85 | 0.775 | 0.242 | 2.9 |

Table 4: Effectiveness of different modules. The first block presents results from a single base model using different mask strategies. The second and third blocks evaluate the EMA-GS and GS-GS setups, respectively. "w/ $M_{h/s}$" indicates alternating between $M_h$ and $M_s$ at each training iteration. "w/o M" denotes that no mask is applied. The results are averaged across scenes within each dataset.

| | Dataset | PhotoTourism | | | NeRF On-the-go | | | RobustNeRF | | |
|---|---|---|---|---|---|---|---|---|---|---|
| Line | Method | PSNR↑ | SSIM↑ | LPIPS↓ | PSNR↑ | SSIM↑ | LPIPS↓ | PSNR↑ | SSIM↑ | LPIPS↓ |
| 1 | Single w/ $M_{h/s}$ | 24.76 | 0.864 | 0.167 | 22.97 | 0.798 | 0.133 | 29.62 | 0.914 | 0.083 |
| 2 | Single w/o M | 24.68 | 0.859 | 0.174 | 19.67 | 0.669 | 0.269 | 25.67 | 0.881 | 0.126 |
| 3 | Single w/o $M_s$ | 24.82 | 0.862 | 0.172 | 22.97 | 0.802 | 0.125 | 28.97 | 0.908 | 0.088 |
| 4 | Single w/o $M_h$ | 24.83 | 0.862 | 0.171 | 22.24 | 0.783 | 0.153 | 29.38 | 0.912 | 0.086 |
| 5 | EMA-GS | 25.24 | 0.864 | 0.169 | 23.40 | 0.801 | 0.135 | 30.27 | 0.917 | 0.080 |
| 6 | EMA-GS w/o M | 24.70 | 0.862 | 0.170 | 21.68 | 0.766 | 0.172 | 28.42 | 0.905 | 0.092 |
| 7 | EMA-GS w/o $M_s$ | 24.76 | 0.864 | 0.166 | 22.94 | 0.802 | 0.126 | 29.08 | 0.909 | 0.089 |
| 8 | EMA-GS w/o $M_h$ | 24.84 | 0.863 | 0.169 | 22.15 | 0.780 | 0.159 | 29.29 | 0.911 | 0.087 |
| 9 | EMA-GS w/o $\mathcal{L}_m$ | 24.73 | 0.861 | 0.174 | 23.10 | 0.801 | 0.132 | 29.72 | 0.915 | 0.081 |
| 10 | EMA-GS w/ Mixup | 25.13 | 0.866 | 0.162 | 23.39 | 0.802 | 0.132 | 30.18 | 0.915 | 0.081 |
| 11 | EMA-GS w/ Dropout | 25.11 | 0.863 | 0.171 | 23.39 | 0.804 | 0.130 | 30.19 | 0.913 | 0.082 |
| 12 | GS-GS | 25.62 | 0.872 | 0.150 | 23.61 | 0.809 | 0.143 | 30.61 | 0.921 | 0.080 |
| 13 | GS-GS w/o M | 25.09 | 0.867 | 0.160 | 22.39 | 0.785 | 0.164 | 29.45 | 0.914 | 0.087 |
| 14 | GS-GS w/o $M_s$ | 25.24 | 0.867 | 0.159 | 23.32 | 0.812 | 0.132 | 30.38 | 0.920 | 0.079 |
| 15 | GS-GS w/o $M_h$ | 25.11 | 0.867 | 0.161 | 22.76 | 0.794 | 0.157 | 30.17 | 0.919 | 0.082 |
| 16 | GS-GS w/o $\mathcal{L}_m$ | 25.08 | 0.869 | 0.155 | 23.13 | 0.808 | 0.135 | 29.92 | 0.918 | 0.082 |

**EMA proxy.** The effectiveness of the EMA-GS setup heavily depends on the masking strategy (L5 *vs.* L7 & L8). Using only a single type of mask often leads to performance similar to or even worse than the single-model baseline (L3 *vs.* L7, L4 *vs.* L8), indicating that confirmation bias can undermine robustness. This highlights the need for diverse masking signals to fully exploit the benefits of the EMA proxy. Moreover, as introduced in the Alternating Masking Strategy part of Section 3.4, we have also tried other forms of regularization to improve the performance of our EMA model, such as random mixup or dropout. However, as shown in L10 and L11 of Table 4, these methods do work as good as our original approach of alternating between the two masking strategies.

## 5 Conclusion

In this work, we present AsymGS, a robust and efficient framework for 3D scene reconstruction in unconstrained, in-the-wild environments. Our method employs two 3DGS models guided by distinct masking strategies to enforce cross-model consistency, effectively mitigating artifacts caused by low-quality observations. To further improve training efficiency, we introduce a dynamic EMA proxy that significantly reduces computational cost with minimal impact on performance. Extensive experiments on three challenging real-world datasets validate the effectiveness and generality of our approach.

## Acknowledgments and Disclosure of Funding

This research was supported by the National Natural Science Foundation of China (62302385). Chengqi Li and Wenbo He thank the NSERC Discovery grant on Towards a Robust and Trustworthy Machine Learning for Visual Contents in Data-centric Applications. Chengqi Li also gratefully acknowledges the Neural Information Processing Foundation for the NeurIPS 2025 Scholar Award supporting his travel.

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

Figure 6: Randomness of artifacts across training runs. Row 1 shows the target view (with the presence of distractors); Rows 2 and 3 present results from two independent runs of Mip-Splatting; Row 4 shows the result of our method with the mutual consistency regularization.

## A    Random nature of artifacts

As shown in Figure 1 and 6, different runs of 3DGS on the same scene (with only the view order randomized) result in different artifacts, particularly in uncertain regions. The mutual consistency loss helps suppress these artifacts in both models. On one hand, the shared static regions remain consistent and act as a strong regularizer. On the other hand, the differing artifacts in uncertain areas provide complementary supervision signals, allowing regions affected by artifacts in one model to be recovered by the other.

For the effect of distinct masking strategies, Table 4 in the main paper presents a quantitative comparison. The performance degrades when both models use the same mask, for both GS-GS and EMA-GS settings. This supports our claim that using separate masks helps prevent convergence to the same erroneous reconstruction patterns.

## B    Comparison with HybridGS

Our Asymmetric Dual 3DGS framework differs fundamentally from HybridGS [13] in both design and training. HybridGS separates static and dynamic content using two models (3DGS for static, 2DGS for dynamic) and requires a staged training process with a learnable blending mask. In contrast, our method uses a dual 3DGS setup with mutual supervision to improve robustness against dynamic noise, all within the standard 3DGS training pipeline. While both methods use masking, HybridGS blends outputs based on 2DGS-derived uncertainty, whereas we apply two distinct masking strategies to reduce confirmation bias from a single, potentially inaccurate mask.

## C    Multi-cue adaptive mask

Some prior works [19, 18] use residuals between ground-truth and rendered images to detect distractors, assuming static regions are learned first. However, this can misclassify object boundaries and miss distractors resembling the background. Others [14, 17] use pretrained semantic segmentation to mask known distractors, such as people or sky, but these methods rely on task-specific priors and lack generality across diverse scenes. We propose Multi-Cue Adaptive Masking to combine the strengths

of residual-based and segmentation-based methods, while also providing a complementary hard mask that captures distinct error patterns compared to the self-supervised soft mask.

---

**Algorithm 1** Multi-Cue Adaptive Masking

---

**Require:** Rendered image $\tilde{\mathbf{I}}$, ground-truth image $\mathbf{I}$, semantic masks $\{\mathbf{M}_k\}$ from SAM, stereo correspondence map $\mathbf{S}_{\geq 3}$ from COLMAP

1:  $\mathbf{E}_{\mathrm{pix}} = \|\tilde{\mathbf{I}} - \mathbf{I}\|_1$                                     ▷ Pixel-level residual
2:  $\mathbf{F} = \mathrm{DINOv2}(\mathbf{I}); \tilde{\mathbf{F}} = \mathrm{DINOv2}(\tilde{\mathbf{I}})$                            ▷ DINOv2 features
3:  $\mathbf{E}_{\mathrm{feat}} = 1 - \mathrm{CosineSimilarity}(\tilde{\mathbf{F}}, \mathbf{F})$                    ▷ Feature-level residual
4:  $\bar{e}_{\mathrm{pix}} = \sum \mathbf{E}_{\mathrm{pix}}/\mathrm{Area}(\mathbf{I})$                         ▷ Average residuals over $\mathbf{I}$
5:  $\bar{e}_{\mathrm{feat}} = \sum \mathbf{E}_{\mathrm{feat}}/\mathrm{Area}(\mathbf{I})$
6:  $\bar{s} = \sum \mathbf{S}_{\geq 3}/\mathrm{Area}(\mathbf{I})$                         ▷ Stereo correspondence density over $\mathbf{I}$
7:  **for** each mask $\mathbf{M}_k$ **do**
8:      $e_{\mathrm{pix},k} = \sum \mathbf{M}_k \odot \mathbf{E}_{\mathrm{pix}} / \sum \mathbf{M}_k$               ▷ Average residuals over $\mathbf{M}_k$
9:      $e_{\mathrm{feat},k} = \sum \mathbf{M}_k \odot \mathbf{E}_{\mathrm{feat}} / \sum \mathbf{M}_k$
10:    $s = \sum \mathbf{M}_k \odot \mathbf{S}_{\geq 3} / \sum \mathbf{M}_k$            ▷ Stereo correspondence density over $\mathbf{M}_k$
11:    **if** $e_{\mathrm{pix},k} > \bar{e}_{\mathrm{pix}}$ and $e_{\mathrm{feat},k} > \bar{e}_{\mathrm{feat}}$ and $s < 0.1 \cdot \bar{s}$ **then**
12:        Mark $\mathbf{M}_k$ as a distractor mask
13:    **end if**
14: **end for**
15: **return** $\mathbf{M}_h = 1 - \bigcup\{\mathbf{M}_k\}_{\mathrm{selected}}$                   ▷ 0 for distractor

---

Here, the stereo-based correspondence records the number of matches each pixel in the given image has, based on SIFT feature correspondences proposed in COLMAP [21]. A pixel is considered a valid correspondence (with the stereo correspondence map value set to true at the pixel location) if its match count exceeds a threshold, indicating it likely belongs to a static region. In contrast, distractors typically yield fewer matches due to their limited presence across images. In Algorithm 1, $\mathbf{S}_{\geq 3}$ denotes the stereo correspondence map, where a pixel is considered a valid correspondence if it has more than three matches.

## D   Datasets and metrics

We evaluate our method on three in-the-wild datasets with varying challenges, as shown in Table 5. NeRF On-the-go dataset [18] features indoor and outdoor sequences with consistent appearance but varying distractor ratios (5%–30%). RobustNeRF dataset [19] provides indoor scenes with static geometry and controlled distractor placement (from single-type to 150 varied distractors), where training is done on cluttered views and testing on clean, unseen ones. We use the undistorted versions of these datasets, following the protocols of WildGaussian [11] and HybridGS [13]. The PhotoTourism dataset [6] includes landmark scenes (Brandenburg Gate, Sacre Coeur, Trevi Fountain) captured under diverse lighting, weather, and viewpoints, with both significant appearance variation and real-world distractors. We report PSNR, SSIM, and LPIPS [27] to assess reconstruction accuracy and perceptual quality.

## E   Implementation details

Our base model is built on Mip-Splatting [25]. Following its default settings, we recompute the sampling rate of each Gaussian every 100 iterations, with a 2D Mip filter variance of 0.1 and a 3D smoothing filter variance of 0.2. We train for 30,000 iterations on NeRF On-the-go and RobustNeRF, with densification and pruning every 1,000 steps until iteration 15,000; and for 100,000 iterations on PhotoTourism, with densification and pruning every 1,000 steps until iteration 50,000. We omit the opacity reset and apply a 1,000-step warm-up before the mutual consistency regularization begins. The consistency regularization weight is set to 0.1. The learnable mask is optimized by a loss weighted $\lambda_{\mathrm{mask}} = 1.0$ with a learning rate of 0.1. For EMA, we use a smoothing factor of $\beta = 0.8$. Semantic regions for the multi-cue adaptive mask are generated using Semantic SAM [12] to create instance-level segmentations and apply Algorithm 1 to select distractor regions as masks.

Table 5: In-the-wild 3D reconstruction datasets.

| Dataset | Scene | # Train | # Test | Distractor | Appear. change |
|---------|-------|---------|--------|------------|----------------|
| NeRF On-the-go [18] | Patio-high | 222 | 45 | ∼30% | No |
| | Spot | 168 | 10 | ∼30% | No |
| | Patio | 98 | 26 | 15%∼20% | No |
| | Corner | 101 | 20 | 15%∼20% | No |
| | Fountain | 168 | 17 | 5%∼10% | No |
| | Mountain | 119 | 12 | 5%∼10% | No |
| RobustNeRF [19] | Statue | 255 | 19 | 1 type | No |
| | Android | 122 | 19 | 1 type | No |
| | Yoda | 109 | 202 | 100 types | No |
| | Crab | 109 | 194 | 150 types | No |
| PhotoTourism [6] | Brandenburg Gate | 763 | 10 | ∼3.5% | Yes |
| | Sacre Coeur | 830 | 21 | ∼3.5% | Yes |
| | Trevi Fountain | 1689 | 19 | ∼3.5% | Yes |

Additionally, we use a 32-dimensional per-view appearance embedding and a 24-dimensional per-Gaussian embedding. Color transformation is performed using a three-layer MLP with hidden size 128, outputting a scale and bias for each RGB channel. The learning rates are set to 0.001 for the per-view embedding, 0.005 for the per-Gaussian embedding, and 0.0005 for the MLP. The other 3DGS-related hyperparameters follow the setup from origin work shown in Table 6.

Table 6: The other 3DGS-related hyperparameters.

| Parameter | Value |
|-----------|-------|
| position_lr_init | 0.00016 |
| position_lr_final | 0.0000016 |
| position_lr_delay_mult | 0.01 |
| feature_lr | 0.0025 |
| opacity_lr | 0.1 |
| scaling_lr | 0.005 |
| rotation_lr | 0.001 |
| percent_dense | 0.01 |
| lambda_dssim | 0.2 |
| densification_interval | 1000 |
| opacity_reset_interval | No opacity reset |
| densify_from_iter | 500 |
| densify_grad_threshold | 0.0002 |

Table 7: The code repo and licenses.

| Method | Link | License |
|--------|------|---------|
| 3DGS [8] | https://github.com/graphdeco-inria/gaussian-splatting | Custom |
| Mip-Splatting [25] | https://github.com/autonomousvision/mip-splatting | Custom |
| WildGaussians [11] | https://github.com/jkulhanek/wild-gaussians/ | MIT License |
| NerfBaselines [10] | https://github.com/nerfbaselines/nerfbaselines | MIT License |
| COLMAP [21] | https://github.com/colmap/colmap | BSD License |
| Semantic-SAM [12] | https://github.com/UX-Decoder/Semantic-SAM | Apache 2.0 License |
| NeRF On-the-go dataset [18] | https://github.com/cvg/nerf-on-the-go | Apache 2.0 License |
| RobustNeRF dataset [19] | https://robustnerf.github.io/ | Custom |
| PhotoTourism dataset [6] | https://github.com/ubc-vision/image-matching-benchmark | Apache 2.0 License |

Table 8: Quantitative results on the NeRF On-the-go dataset [18]. The best and second-best results are highlighted in **bold** and underline, respectively.

| Scene | Mountain | | | Fountain | | | Corner | | | Patio | | | Spot | | | Patio-High | | |
|---|---|---|---|---|---|---|---|---|---|---|---|---|---|---|---|---|---|---|
| Method | PSNR↑ | SSIM↑ | LPIPS↓ | PSNR↑ | SSIM↑ | LPIPS↓ | PSNR↑ | SSIM↑ | LPIPS↓ | PSNR↑ | SSIM↑ | LPIPS↓ | PSNR↑ | SSIM↑ | LPIPS↓ | PSNR↑ | SSIM↑ | LPIPS↓ |
| RobustNeRF [19] | 17.54 | 0.496 | 0.383 | 15.65 | 0.318 | 0.576 | 23.04 | 0.764 | 0.244 | 20.39 | 0.718 | 0.251 | 20.65 | 0.625 | 0.391 | 20.54 | 0.578 | 0.366 |
| NeRF On-the-go [18] | 20.15 | 0.644 | 0.259 | 20.11 | 0.609 | 0.314 | 24.22 | 0.806 | 0.190 | 20.78 | 0.754 | 0.219 | 23.33 | 0.787 | 0.189 | 21.41 | 0.718 | 0.235 |
| 3DGS [8] | 19.40 | 0.638 | 0.213 | 19.96 | 0.659 | 0.185 | 20.90 | 0.713 | 0.241 | 17.48 | 0.704 | 0.199 | 20.77 | 0.693 | 0.316 | 17.29 | 0.604 | 0.363 |
| Mip-Splatting [25] | 19.86 | 0.649 | 0.200 | 20.19 | 0.672 | 0.189 | 21.15 | 0.728 | 0.230 | 18.31 | 0.639 | 0.328 | 20.18 | 0.689 | 0.338 | 18.31 | 0.639 | 0.328 |
| WildGaussian [11] | 20.43 | 0.653 | 0.255 | 20.81 | 0.662 | 0.215 | 24.16 | 0.822 | **0.045** | 21.44 | 0.800 | 0.138 | 23.82 | 0.816 | 0.138 | 22.23 | 0.725 | 0.206 |
| SLS-mlp [20] | 19.84 | 0.580 | 0.294 | 20.19 | 0.612 | 0.258 | 24.03 | 0.795 | 0.258 | 21.55 | 0.838 | **0.065** | 23.52 | 0.756 | 0.185 | 20.31 | 0.664 | 0.259 |
| HybridGS [13] | 21.73 | 0.693 | 0.284 | 21.11 | 0.674 | 0.252 | 25.03 | 0.847 | 0.151 | 21.98 | 0.812 | 0.169 | 24.33 | 0.794 | 0.196 | 21.77 | 0.741 | 0.211 |
| Ours (GS-GS) | **22.00** | **0.740** | 0.199 | **21.83** | **0.717** | 0.180 | **26.15** | **0.885** | 0.085 | **22.97** | **0.860** | 0.096 | **25.52** | **0.854** | **0.135** | **23.17** | 0.796 | 0.164 |
| Ours (EMA-GS) | 21.93 | 0.735 | **0.162** | 21.61 | 0.709 | **0.162** | 25.77 | 0.876 | 0.089 | 22.87 | 0.853 | 0.091 | 25.09 | 0.839 | 0.152 | 23.14 | **0.797** | **0.156** |

# F More results

## F.1 NeRF On-the-go and RobustNeRF

In Table 1 and Table 8, our method (GS-GS) outperforms all baseline methods by more than 1 dB in scenes with medium to high occlusion ratios. The margin is smaller in low-occlusion scenes, where 3DGS-based methods already perform well due to strong geometric priors from the initial point cloud. A similar trend is observed in Table 2: while the proposed method surpasses the SOTA by approximately 0.4 dB in simpler scenes containing a single distractor type (e.g., Statue and Android), it outperforms others by more than 1 dB in complex scenes with a large number of diverse distractors (e.g., Yoda and Crab). The rendering results in Figure 7 and 8 further demonstrate the superiority of our method, as competing approaches exhibit distractor remains and missing details.

## F.2 PhotoTourism

The Asymmetric Dual 3DGS achieves an average improvement of 0.8 dB on the PhotoTourism dataset (Table 3), demonstrating its effectiveness under challenging appearance variations. Furthermore, proper appearance modeling is essential for handling in-the-wild data with diverse visual conditions. This is supported by a significant performance gap of more than 4 dB between methods with and without appearance modeling, as shown in Table 3, and further illustrated by the visual differences in Figure 9. Therefore, we apply appearance modeling for the PhotoTourism dataset by default. As the importance of appearance modeling is addressed here, we omit further discussion in the following ablation section and apply appearance modeling by default for the PhotoTourism dataset.

## F.3 Statistical significance of the main result

Table 9: Quantitative results on the NeRF On-the-go dataset. Each experiment is repeated five times, and we report the mean and standard deviation.

| Setting | GS-GS | | | EMA-GS | | |
|---|---|---|---|---|---|---|
| Scene | PSNR | SSIM | LPIPS | PSNR | SSIM | LPIPS |
| High Occlusion | $24.36 \pm 0.02$ | $0.823 \pm 0.001$ | $0.151 \pm 0.001$ | $24.11 \pm 0.05$ | $0.819 \pm 0.002$ | $0.152 \pm 0.004$ |
| Medium Occlusion | $24.52 \pm 0.06$ | $0.871 \pm 0.001$ | $0.090 \pm 0.001$ | $24.26 \pm 0.08$ | $0.864 \pm 0.001$ | $0.092 \pm 0.002$ |
| Low Occlusion | $21.99 \pm 0.04$ | $0.730 \pm 0.001$ | $0.184 \pm 0.004$ | $21.81 \pm 0.09$ | $0.723 \pm 0.002$ | $0.166 \pm 0.007$ |

Table 10: Quantitative results on the RobustNeRF dataset. Each experiment is repeated five times, and we report the mean and standard deviation.

| Setting | GS-GS | | | EMA-GS | | |
|---|---|---|---|---|---|---|
| Scene | PSNR | SSIM | LPIPS | PSNR | SSIM | LPIPS |
| Statue | $23.44 \pm 0.05$ | $0.893 \pm 0.001$ | $0.098 \pm 0.001$ | $23.46 \pm 0.06$ | $0.890 \pm 0.001$ | $0.097 \pm 0.001$ |
| Android | $25.58 \pm 0.05$ | $0.856 \pm 0.001$ | $0.070 \pm 0.003$ | $25.47 \pm 0.06$ | $0.849 \pm 0.002$ | $0.070 \pm 0.002$ |
| Yoda | $37.12 \pm 0.09$ | $0.969 \pm 0.001$ | $0.074 \pm 0.001$ | $36.46 \pm 0.06$ | $0.967 \pm 0.001$ | $0.078 \pm 0.001$ |
| Crab | $36.11 \pm 0.07$ | $0.963 \pm 0.001$ | $0.079 \pm 0.001$ | $35.52 \pm 0.07$ | $0.961 \pm 0.001$ | $0.080 \pm 0.001$ |

We repeated the experiment five times. Based on the results in Table 9 and 10, our method shows statistically significant improvements.

### F.4 Hyperparameters

We perform hyperparameter tuning on the NeRF On-the-go dataset [18] to optimize the performance of our method (GS-GS and EMA-GS). As shown in Table 11, we tune the EMA smoothing factor $\beta$ and find that $\beta = 0.8$ yields the highest PSNR and SSIM with the lowest LPIPS. In Table 12, we evaluate different densification intervals and observe that an interval of 1000 offers the best overall performance. Similarly, Table 13 presents the results of tuning the warm-up interval, where 1000 again emerges as the optimal choice, outperforming both shorter and longer intervals. Lastly, Table 14 shows that removing opacity reset improves reconstruction quality, suggesting that preserving learned opacity leads to more stable and effective training.

Table 11: Tuning the EMA smoothing factor according to the average performance on the NeRF On-the-go dataset [18].

| $\beta$ | PSNR↑ | SSIM↑ | LPIPS↓ |
|---|---|---|---|
| 0.5 | 22.80 | 0.797 | 0.136 |
| 0.6 | 22.93 | 0.797 | 0.137 |
| 0.7 | 23.12 | 0.799 | 0.136 |
| 0.8 | 23.40 | 0.801 | 0.135 |
| 0.9 | 23.05 | 0.798 | 0.136 |

Table 12: Tuning the densification interval according to the average performance on the NeRF On-the-go dataset [18].

| Setting | GS-GS | | | EMA-GS | | |
|---|---|---|---|---|---|---|
| Densification Interval | PSNR↑ | SSIM↑ | LPIPS↓ | PSNR↑ | SSIM↑ | LPIPS↓ |
| 500 | 23.60 | 0.810 | 0.129 | 23.00 | 0.797 | 0.134 |
| 1000 | 23.61 | 0.810 | 0.135 | 23.40 | 0.801 | 0.135 |
| 1500 | 23.58 | 0.807 | 0.146 | 23.15 | 0.796 | 0.143 |
| 2000 | 23.56 | 0.806 | 0.152 | 22.96 | 0.797 | 0.145 |

Table 13: Tuning the warm-up interval according to the average performance on the NeRF On-the-go dataset [18].

| Setting | GS-GS | | | EMA-GS | | |
|---|---|---|---|---|---|---|
| Warm-up Interval | PSNR↑ | SSIM↑ | LPIPS↓ | PSNR↑ | SSIM↑ | LPIPS↓ |
| 0 | 23.55 | 0.808 | 0.137 | 22.96 | 0.798 | 0.135 |
| 500 | 23.55 | 0.809 | 0.137 | 23.08 | 0.799 | 0.134 |
| 1000 | 23.61 | 0.810 | 0.135 | 23.40 | 0.801 | 0.135 |
| 1500 | 23.58 | 0.809 | 0.137 | 23.10 | 0.799 | 0.135 |
| 2000 | 23.60 | 0.810 | 0.135 | 22.88 | 0.798 | 0.136 |

In Table 15 and 16, although the best performance is generally achieved at our default setting ($\lambda_m = 1.0$ and $\lambda_{mask} = 1.0$ for GS-GS; $\lambda_m = 0.1$ and $\lambda_{mask} = 1.0$ for EMA-GS), the differences across settings are minimal (less than 0.1 dB). This indicates that the performance is not highly sensitive to the values of $\lambda_m$ and $\lambda_{mask}$.

## G   Limitations

We adopt the appearance modeling approach from WildGaussian [11], using a per-view appearance embedding to control global appearance and a per-Gaussian embedding to model the appearance of individual Gaussian primitives. However, this model struggles to capture fine-grained effects such as object highlights. A likely reason is the limited diversity in training data. To address this, we plan to introduce data augmentation with randomized illumination variations.

Table 14: Impact of opacity reset on reconstruction quality, evaluated on the NeRF On-the-go dataset [18].

| Setting | GS-GS | | | EMA-GS | | |
|---|---|---|---|---|---|---|
| Opacity Reset | PSNR↑ | SSIM↑ | LPIPS↓ | PSNR↑ | SSIM↑ | LPIPS↓ |
| w/o | 23.61 | 0.810 | 0.135 | 23.40 | 0.801 | 0.135 |
| w/ | 22.87 | 0.790 | 0.176 | 22.43 | 0.786 | 0.158 |

Table 15: Performance under varying weights of the mutual consistency loss, evaluated on the NeRF On-the-go dataset [18].

| | GS-GS | | | | EMA-GS | | |
|---|---|---|---|---|---|---|---|
| $\lambda_m$ | PSNR | SSIM | LPIPS | $\lambda_m$ | PSNR | SSIM | LPIPS |
| 0.0 | 23.13 | 0.808 | 0.135 | 0.0 | 23.10 | 0.801 | 0.132 |
| 0.5 | 23.66 | 0.810 | 0.130 | 0.05 | 23.39 | 0.803 | 0.136 |
| 1.0 | 23.61 | 0.810 | 0.135 | 0.1 | 23.40 | 0.801 | 0.135 |
| 1.5 | 23.54 | 0.807 | 0.142 | 0.2 | 23.43 | 0.805 | 0.133 |
| 2.0 | 23.44 | 0.803 | 0.149 | 0.3 | 23.47 | 0.805 | 0.134 |

Table 16: Performance under varying weights of the learnable mask loss, evaluated on the NeRF On-the-go dataset [18].

| Setting | GS-GS | | | EMA-GS | | |
|---|---|---|---|---|---|---|
| $\lambda_{mask}$ | PSNR | SSIM | LPIPS | PSNR | SSIM | LPIPS |
| 0.5 | 23.62 | 0.809 | 0.136 | 23.33 | 0.802 | 0.134 |
| 1.0 | 23.61 | 0.810 | 0.135 | 23.40 | 0.801 | 0.135 |
| 1.5 | 23.59 | 0.809 | 0.137 | 23.41 | 0.804 | 0.135 |
| 2.0 | 23.63 | 0.811 | 0.135 | 23.32 | 0.802 | 0.134 |

## H Social impact

Notre-Dame de Paris suffered a devastating fire in 2019. Although the building was severely damaged, restoration was aided by a 3D model originally created for a video game, highlighting the importance of preserving 3D models of cultural landmarks. However, such sites are often crowded with people, and photos taken at different times may exhibit varying lighting conditions. This highlights the broader societal benefit of accessible and robust 3D scene reconstruction technologies. Our method contributes positively by enabling the creation of high-quality 3D models from in-the-wild images, which are often affected by distractors and lighting variations. By making it feasible to reconstruct cultural landmarks from everyday photos, our approach supports digital preservation, education, and historical restoration efforts.

There are potential negative impacts, such as misuse in surveillance or privacy-invading applications. In particular, in-the-wild image collections often contain individuals who are unintentionally captured. To mitigate this risk, we recommend removing or anonymizing identifiable information, such as faces or bodies, from the reconstructed scenes. This can be achieved through automated segmentation or masking techniques applied before or during training.

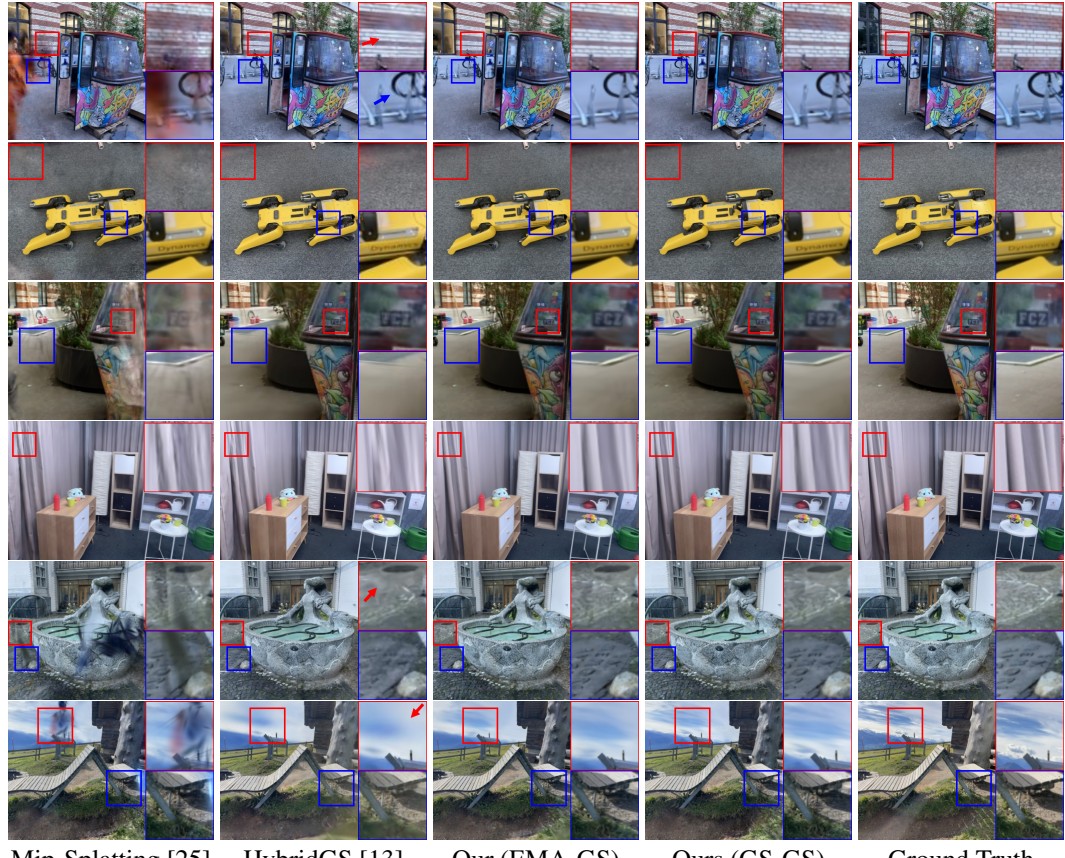

Mip-Splatting [25]     HybridGS [13]     Our (EMA-GS)     Ours (GS-GS)     Ground Truth

Figure 7: Qualitative results on the NeRF On-the-go dataset [18]. The scenes shown are, from top to bottom: Patio-high (high occlusion), Spot (high occlusion), Patio (medium occlusion), Corner (medium occlusion), Mountain (low occlusion), and Fountain (low occlusion).

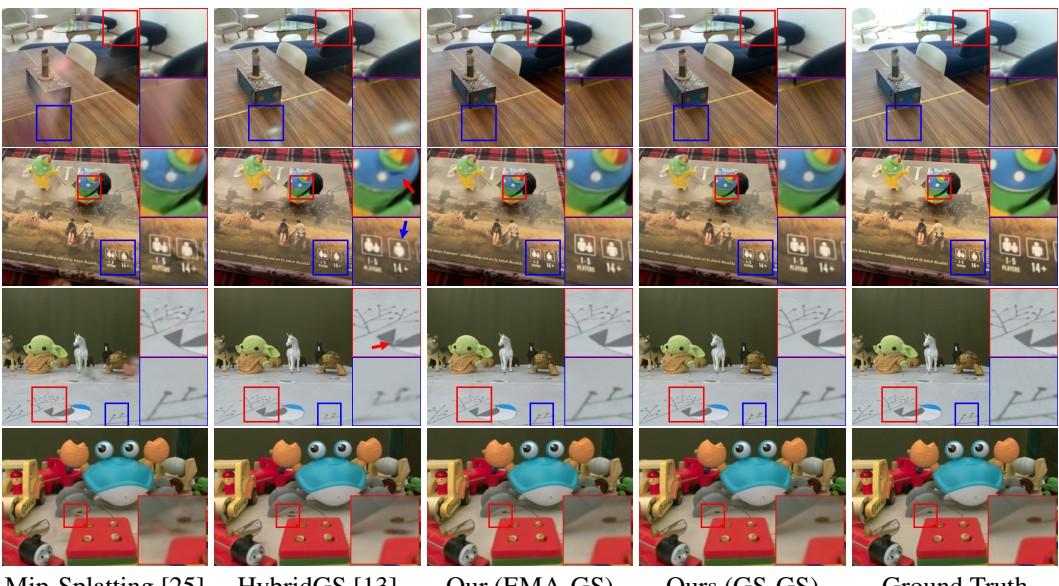

Mip-Splatting [25]     HybridGS [13]     Our (EMA-GS)     Ours (GS-GS)     Ground Truth

Figure 8: Qualitative results on the RobustNeRF dataset [19]. The scenes shown are, from top to bottom: Statue, Android, Yoda, and Crab.

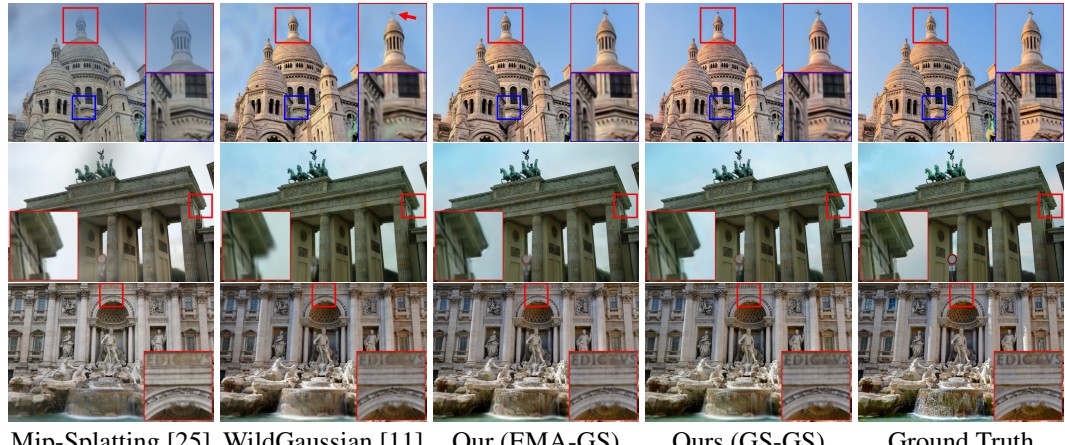

| Mip-Splatting [25] | WildGaussian [11] | Our (EMA-GS) | Ours (GS-GS) | Ground Truth |

Figure 9: Qualitative results on the PhotoTourism dataset [6]. The scenes shown are, from top to bottom: Sacre Coeur, Brandenburg Gate, and Trevi Fountain.

