# OpenReview forum: "Robust Neural Rendering in the Wild with Asymmetric Dual 3D Gaussian Splatting"
_NeurIPS.cc/2025/Conference — NeurIPS 2025 spotlight_

### Official Review · Reviewer_8u6B · 2025-06-02

**Clarity:** 3
**Significance:** 3
**Originality:** 4
**Rating:** 5
**Confidence:** 5

**Summary:**

This paper propose a novel and interesting framework to tackle distractors for in-the-wild scene reconstruction. The authors leverage the stochastic nature of artifacts caused by distractors to form a mutual consistency loss for different runs. To avoid both runs converage toward the same reconstruction errors, the authors apply two types of dynamic masking strategy for two models. For computational efficiency, the authors apply EMA strategy during optimization to avoid double gradient descent. In a nutshell, this paper is interesting and somewhat intuitive.

**Questions:**

For Eq.(7), will it still work in the case that the reconstructed 3D scene overfits to the distractors? If not, how does this method avoid overfitting to distractors with 3D Gaussians floating in the sky?
More can be seen in "Weakness"

**Ethical Concerns:**

["NO or VERY MINOR ethics concerns only"]

**Final Justification:**

I've read all the reviewers' questions and the authors' responses. The authors have addressed most of the reviewers' questions. No further questions. Because most reviewers concern about the relationship between the mask strategy and hybrid-GS, the authors should add more visual comparisons and examples in the camera-ready version / supplementary material according to the reviewers' suggestions.

**Limitations:**

yes

**Quality:**

3

**Strengths And Weaknesses:**

**Strengths:**

1. This paper propose to leverage the stochastic nature of artifacts to tackle distractors, which is insteresting.
2. To obtain mutual consistency across different runs, this paper trains two models in parallel, and further accelerate the training speed with EMA strategy.
3. The ablation study is comprehensive and convincing.

**Weakness:**

1. My major concern is that some statements in this paper requires further visual analysis to support, especially the stochastic nature of artifacts, which is the core idea in this paper.
    - For example, I wanna see more examples across various scenes and different runs to demonstrate the appearance of artifcats is stochastic.
    - The visual comparison between applying the same mask or different mask on two parallel models is also necessary to show whether this design can avoid converging toward the same reconstruction errors
    - Does the above conclusion also work for EMA stategy? A visual comparison is also needed here.

---

> ### Author Rebuttal · Authors · 2025-07-29
>
> Thank you for your thoughtful feedback and valuable suggestions! We are sorry that we cannot include images showing the stochastic artifacts according to new rebuttal rules, but we are committed to including more visual evidences in the next step.
>
> ---
> **W1: My major concern is that some statements in this paper require further visual analysis to support, especially the stochastic nature of artifacts, which is the core idea in this paper. For example, I wanna see more examples across various scenes and different runs to demonstrate the appearance of artifacts is stochastic. The visual comparison between applying the same mask or different masks on two parallel models is also necessary to show whether this design can avoid converging toward the same reconstruction errors. Does the above conclusion also work for EMA strategy? A visual comparison is also needed here.**
>
> We sincerely appreciate the reviewer’s insightful comments. Unfortunately, unlike NeurIPS last year, this venue does not provide a mechanism to submit a rebuttal PDF with visual evidences.
>
> Regarding **the stochastic nature of artifacts**, we included visual examples in the main paper. As shown in Figure 1, different runs of 3DGS on the same scene (with only the view order randomized) result in different artifacts, particularly in uncertain regions. The mutual consistency loss helps suppress these artifacts in both models. On one hand, the shared static regions remain consistent and act as a strong regularizer. On the other hand, the differing artifacts in uncertain areas provide complementary supervision signals, allowing regions affected by artifacts in one model to be recovered by the other.
>
> For **effect of distinct masking strategies**, Table 4 in the main paper presents a quantitative comparison. The performance degrades when both models use the same mask, for both GS-GS and EMA-GS settings. This supports our claim that using separate masks helps prevent convergence to the same erroneous reconstruction patterns.
>
> We agree that linking quantitative results to visual evidence would strengthen our conclusions. Therefore, we are committed to including additional visual comparisons across different scenes, masking strategies, and framework settings in the supplementary materials.
>
> ---
> **Q1: For Eq.(7), will it still work in the case that the reconstructed 3D scene overfits to the distractors? If not, how does this method avoid overfitting to distractors with 3D Gaussians floating in the sky?**
>
> Yes, Eq.(7) remains effective even if the 3D reconstruction overfits to distractors.
>
> When the model overfits to distractors, the rendered regions corresponding to distractors will exhibit noticeable deviations from the ground truth. These discrepancies result in low similarity scores, which serve as targets for the learnable mask as shown in Eq.(7). As a result, the mask is driven toward zero in those regions, effectively suppressing the influence of distractors.
>
> For floating Gaussians in the sky, we adopt the pruning strategy from the original 3DGS implementation to remove transient or oversized primitives, preventing their accumulation in regions without valid geometry.

---

> > ### Comment · Reviewer_8u6B · 2025-08-02
> > **No further questions**
> >
> > I've read all reviewers' questions and the authors' responses. The authors have addressed most of the reviewers' questions. No further questions. Because most reviewers concern the relationship between the mask strategy and hybrid-GS, the authors should add more visual comparisons and examples in the camera-ready version / supplementary material according to the reviewers' suggestions.

---

> > > ### Author Response · Authors · 2025-08-02
> > >
> > > Again, thank you for the valuable feedback. As recommended, we will include additional details on artifacts and comparisons with similar methods in the camera-ready version and supplementary material.

---

### Official Review · Reviewer_b1d5 · 2025-07-01

**Clarity:** 3
**Significance:** 2
**Originality:** 3
**Rating:** 4
**Confidence:** 4

**Summary:**

This paper proposes a new method to **reconstruct scenes despite transient objects** being present in only some of the frames (e.g. a passerby in an outdoor scene). Specifically, Asymmetric Dual 3DGS optimizes two 3DGS in parallel and leverages the consistency between the two reconstructions to improve them. This is achieved in two steps:
* **Enforcing consistency between the models**: their renderings are supervised to be identical.
* **Preventing the two reconstructions from being identical**: the two 3DGS identify transient objects in different ways, the first 3DGS uses a combination of SAM+COLMAP+DINOv2 and residual error, while the second uses the DINOv2 residual error only (i.e. the difference in DINO features between render and ground-truth).
Then, the authors apply both transient strategies to the first model, and replace the second model by a moving average of the first, reducing the computational overhead of their technique.
Both approaches perform well on a number of datasets with transient objects and varying lighting, comparing against numerous methods such as HybridGS and WildGaussians.

**Questions:**

- It is a bit counter-intuitive that the dual-models would perform better than a single model combining both masking strategies (Single w/ Mh/s). What is the interpretation of this result?
- There are many other ways to keep the models different, and using two different losses for the same goal of masking transient objects is not the most intuitive. Have you experimented with other ideas, such as using different initializations or training on different subsets of images?
Minor points and typos:
- The term "frame-independent" is used with and without dash. It also does not exactly convey the meaning intended.
- In the EMA section, it is unclear at the beginning that the main model uses both masking strategies. It should be mentioned early.

**Ethical Concerns:**

["NO or VERY MINOR ethics concerns only"]

**Final Justification:**

After reading the rebuttal and other reviews, most of my concerns have been resolved. Thus, I retain my rating.

**Limitations:**

- yes, limitations are discussed but only in the supplementary material.
Despite their contributions, these methods largely rely on heuristic strategies to suppress
corrupted supervision signals from low-quality training data. [...] In this work, we propose a
principled framework, Asymmetric Dual 3DGS.

**Quality:**

3

**Strengths And Weaknesses:**

+: The overall observation and philosophy behind the approach are sound and interesting: training two similar 3DGS models to be similar, while making sure they remain different.

-: However in practice, the way these reconstructions are kept different is by introducing more losses to perform the same task of transient mask estimation.

-: The paper does not attempt to meaningfully explain how it differs from previous methods, and instead describes the entire literature of transient NeRF/3DGS as "heuristic strategies". This description is very vague and could be used to describe the proposed method too, which is instead called a "principled framework". For example, the estimation of multi-cue adaptive mask is very "heuristic".

-: In fact, some of these works have a fairly similar idea, such as SimpleNeRF which also uses a simpler model to regularize the main one, or HybridGS which uses an additional 2DGS to separate moving elements. To be clear, these methods are not exactly identical, but the paper would benefit from discussing these differences in detail and why they matter.

+: The experiments compare with several methods for radiance field reconstruction that are robust to transient objects. An extensive ablation study is also performed.

---

> ### Author Rebuttal · Authors · 2025-07-30
>
> Thank you for the insightful review. We’re grateful for your acknowledgment of our key ideas and experimental efforts. We will thoroughly respond to the concerns below.
>
> ---
> **W1: However in practice, the way these reconstructions are kept different is by introducing more losses to perform the same task of transient mask estimation.**
>
> We acknowledge the reviewer’s observation. In our design, we use two distinct masking strategies for each branch of the dual 3DGS model. This explicitly enforces asymmetric supervision over dynamic regions, promoting diversity between the models.
>
> At the same time, we apply a mutual consistency loss to encourage both models to agree in the static regions. This loss also enables each model to recover information that may be suppressed in the other due to masking, helping to mitigate confirmation bias arising from potentially inaccurate masking strategies. In other words, areas blocked in one branch can often be reconstructed using guidance from the other avoiding same masking error.
>
> Overall, our approach maintains divergence in dynamic regions while enforcing consistency in static regions, enabling both robustness and completeness. In Eq. (8), the first two terms correspond to reconstruction under separate masks, while the middle terms encode cross-model consistency, supporting static region reinforcement and dynamic region recovery.
>
> ---
> **W2: The paper does not attempt to meaningfully explain how it differs from previous methods, and instead describes the entire literature of transient NeRF/3DGS as "heuristic strategies". This description is very vague and could be used to describe the proposed method too, which is instead called a "principled framework". For example, the estimation of multi-cue adaptive mask is very "heuristic".**
>
> We refer to existing methods as heuristic based on the design choices they make and the lack of formal principles guiding their generalization. For example, RobustNeRF introduces custom loss functions to suppress noisy training signals. However, this is achieved using several hand-crafted rules, for instance, treating residuals as inliers only if they fall below a certain percentile within a certain-size local neighborhood. While effective empirically, such rules are ad hoc and task-specific, potentially compromising stability when conditions vary.
>
> In contrast, our Dual 3DGS framework is grounded in the principle of mutual consistency. It explicitly leverages the assumption that transient artifacts are random across training runs, guiding the training process toward stable and generalizable representations. This principle-driven formulation allows our method to generalize better to diverse scenes.
>
> Nevertheless, we acknowledge that our approach still involves certain heuristic components. To improve clarity, we will revise the manuscript to replace vague terms like ''heuristic'' and ''principled'' with more specific and objective descriptions of the methodological differences.
>
> ---
> **W3: In fact, some of these works have a fairly similar idea, such as SimpleNeRF which also uses a simpler model to regularize the main one, or HybridGS which uses an additional 2DGS to separate moving elements. To be clear, these methods are not exactly identical, but the paper would benefit from discussing these differences in detail and why they matter.**
>
> Thank you for your advice on more discussions about related methods.
>
> Our approach differs fundamentally from HybridGS in both design and training. HybridGS separates static and dynamic content using two models (3DGS for static, 2DGS for dynamic) and requires a staged training process with a learnable blending mask. In contrast, our method uses a dual 3DGS setup with mutual supervision to improve robustness against dynamic noise, all within the standard 3DGS training pipeline. While both methods use masking, HybridGS blends outputs based on 2DGS-derived uncertainty, whereas we apply two distinct masking strategies to reduce confirmation bias from a single, potentially inaccurate mask.
>
> SimpleNeRF focuses on few-shot 3D reconstruction with sparse input views, using multiple augmented NeRF models (parallel and serial) to provide additional supervisory signals. In contrast, our work tackles robust in-the-wild reconstruction by employing two parallel 3DGS models with mutual consistency regularization, which reinforces static regions while suppressing dynamic artifacts through complementary representations. Additionally, SimpleNeRF requires about 22 hours of training per scene, whereas our asymmetric dual 3DGS completes training in under one hour.
>
> We will elaborate more on these differences in the main paper.
>
> ---
> **Q1: It is a bit counter-intuitive that the dual-models would perform better than a single model combining both masking strategies (Single w/ Mh/s). What is the interpretation of this result?**
>
> Both the dual-model settings (lines 5 and 12 in Table 4) and the single model with alternating masks (line 1) utilize two distinct masks. However, a key difference lies in **the mutual consistency loss** used in the dual-model setup. This loss reinforces reliable signals from static regions and enables one model to help recover regions suppressed by the mask in the other.
>
> When the mutual consistency loss is removed (lines 9 and 16 in Table 4), the performance of the dual-model framework drops and approaches that of the single model, effectively becoming a simple ensemble.
>
> ---
> **Q2: There are many other ways to keep the models different, and using two different losses for the same goal of masking transient objects is not the most intuitive. Have you experimented with other ideas, such as using different initializations or training on different subsets of images?**
>
> Yes, we have explored several strategies to introduce diversity between the models. For instance, we experimented with **random dropout**, where a subset of 3DGS primitives is randomly frozen during optimization, and **random mixup**, where the ground-truth image is blended with the rendered output for a given view. These methods are evaluated in **lines 10 and 11 of Table 4** in the main paper. The results show that, although they introduce randomness, their effectiveness is limited compared to the alternating masking strategy.
>
> We considered using different initializations or subsets of views but ultimately chose not to pursue these approaches. Because the original 3DGS paper demonstrated that deviating from COLMAP-based initialization leads to significant performance degradation. Similarly, training with only partial data adversely affects reconstruction quality.
>
> ---
> **Q3: The term "frame-independent" is used with and without dash. It also does not exactly convey the meaning intended.**
>
> Thank you for pointing this out. The use of "frame-independent" in Line 145 is a typographical error, and we intended to write "frame-dependent". In our work, the appearance modeling is actually view-dependent, as we assign a learnable global appearance vector to each input view and jointly optimize it with the rest of the 3DGS model. The term "frame" was intended to mean "view," but we agree that "view-dependent" is a more accurate and consistent term. We will revise the terminology throughout the paper accordingly.
>
> ---
> **Q4: In the EMA section, it is unclear at the beginning that the main model uses both masking strategies. It should be mentioned early.**
>
> In the EMA section, our primary focus was to highlight the computational efficiency introduced by the EMA proxy design, and the masking strategy was not emphasized upfront. Although the alternating masking strategy is described later in the subsection, we agree that providing a brief mention of it earlier in the section would give readers a clearer overview. We will revise the text to introduce the masking strategy at the beginning for better clarity.

---

> > ### Comment · Reviewer_b1d5 · 2025-08-06
> >
> > Dear all,
> >
> > After reading the rebuttal and other reviews, I think most of my concerns have been addressed. Please include the due discussions in the paper to better present the idea in comparison to existing methods that share a similar idea.

---

> > > ### Author Response · Authors · 2025-08-06
> > >
> > > Thank you for your follow-up and thoughtful review. We're glad that most of your concerns have been addressed. We will incorporate the key discussion points into the paper to better emphasize the distinctions from related methods.
> > >
> > > Given your positive assessment, we kindly ask if you would consider raising your score. We sincerely appreciate your time and valuable feedback throughout the review process.

---

> ### Author Response · Authors · 2025-08-05
>
> Dear Reviewer b1d5,
>
> Thank you for your thoughtful review and constructive feedback. As the discussion phase draws to a close, we’d like to check if our responses have fully addressed your concerns. If there are any remaining questions or points that need clarification, please feel free to let us know.
>
> Thank you.

---

### Official Review · Reviewer_REjw · 2025-07-03

**Clarity:** 4
**Significance:** 3
**Originality:** 3
**Rating:** 5
**Confidence:** 4

**Summary:**

This paper presents a novel method for outlier suppression in 3DGS training. Their method is motivated by the observation that training the same model twice, but with different sample ordering, results in output stochasticity in uncertain areas. They therefore train two model simultaneously enforcing a cross-consistency loss that encourages model to agree photometrically when rendered from the same view. Furthermore, they give each model separate masking strategies to mitigate confirmation bias. These masking strategies are similar to some previous works, but allow them to combine strengths of each. The method also involves some extra machinery to account for appearance modeling and offers an EMA variant offers efficiency gains. The method achieves state of the art results over most datasets and the design choices are thoroughly ablated.

**Questions:**

1. Can you elaborate on the technique of the multi-cue adaptive mask? I'm a bit confused - are "static" regions always included? note that semantically static objects could still be considered as outliers due to difficult to model specular reflections. Are then only the "dynamic" regions considered for the union of the other cues?

2. How is the self-supervised mask regularized against collapsing to zero?

**Ethical Concerns:**

["NO or VERY MINOR ethics concerns only"]

**Final Justification:**

I thank the authors for the information provided in their rebuttal. In my opinion the methods in the "Multi-Cue Adaptive Mask" deserve a bit more attention in the main text. I can't fault the authors for a simple empirical observation about training image ordering. The exact underlying causes of this I'm sure will be answered by further work. I also thank the authors for their explanation about why the self-supervised mask can't collapse like other strategies I've seen.

**Limitations:**

Yes.

**Paper Formatting Concerns:**

None that I could see.

**Quality:**

4

**Strengths And Weaknesses:**

The paper is generally well written in all aspects, and the technique is sufficiently described.

The method, though it is somewhat complicated in it's complete implementation, it achieves state of the art results over a sufficient set of the most relevant datasets, making it a significant contribution. This perhaps isn't too surprising, as their method allows them to combine the strengths of previous methods through the use of different masking strategies in the models. All design choices are thoroughly ablated, justifying the architecture.

Details are scant in the description of the Multi-Cue Adaptive Mask. I would prefer to see some of this in mathematical notation. The method also sounds somewhat similar to the newly released RoMo [1], in that it combines SAM features with epipolar cues.

If I further wanted to nitpick at weaknesses, one could say that because the method could be seen as one that combines strengths of previous works, it could be classed as less than original. The authors observation that retrained models tend to disagree in areas of uncertainty is simply empirical and could have been better motivated. Yet this is difficult to do in a limited amount of space.

It could also be faulted for being somewhat complex - yet there is no other simpler technique that can achieve the same metrics.

[1] https://romosfm.github.io/

---

> ### Author Rebuttal · Authors · 2025-07-30
>
> Thank you for the thoughtful and encouraging review. We truly appreciate your recognition of the paper’s clarity, technical depth, and contributions, as well as your constructive feedback on areas for improvement.
>
> ---
> **W1: Details are scant in the description of the Multi-Cue Adaptive Mask.**
>
> Due to space constraints in the main paper and our focus on presenting the dual-model framework and its efficient variant, we chose to include the detailed formulation of the Multi-Cue Adaptive Mask in the supplementary material. For better clarity, we also provide an expanded explanation in our response to Question 1 below.
>
> ---
> **W2: The authors observation that retrained models tend to disagree in areas of uncertainty is simply empirical and could have been better motivated.**
>
> We observe that randomness in the training process, particularly in the order in which views are fed into the 3DGS optimization, can lead to stochastic visual artifacts. While our paper does not provide a formal theoretical justification for this phenomenon, the underlying intuition is reasonable: if a single noisy view is introduced late in training, its influence may not be sufficiently corrected and could result in more prominent artifacts. Conversely, when introduced earlier, the model may better learn to ignore or compensate for the noise. These inconsistencies motivate our design of a dual 3DGS framework, where two models supervise each other. Since they are trained on the same scene, their shared static regions remain consistent, while the differing artifacts in uncertain areas provide complementary supervision signals, allowing regions affected by artifacts in one model to be recovered by the other.
>
> ---
> **Q1: Can you elaborate on the technique of the multi-cue adaptive mask? I'm a bit confused - are "static" regions always included? note that semantically static objects could still be considered as outliers due to difficult to model specular reflections. Are then only the "dynamic" regions considered for the union of the other cues?**
>
> **(1) Multi-Cue Adaptive Mask**
>
> The Multi-Cue Adaptive Mask is a hard binary mask derived from four cues: semantic segmentation, stereo correspondence, pixel-level residuals, and feature-level residuals.
>
> We first segment each image into **semantic regions** using Semantic-SAM [1], which serve as units for aggregating cues.
>
> For **stereo correspondence**, we use SIFT matches from COLMAP [2]. A pixel with matches in more than three views is considered a reliable static point ($\mathbf{S}_{\ge 3}$ in Algorithm 1 of the supplementary material). Regions with many such points are unlikely to be distractors.
> We also compute **pixel-level** ($L_1$ loss) and **feature-level** (DINOv2 cosine distance) residuals between rendered and ground-truth images. High residuals indicate distractors.
>
> Each region is classified as a distractor if: (1) Its average residual exceeds the global mean, and (2) Its static point density is below the global mean. Distractor regions are excluded by assigning a mask value of zero. **We take the union of all such regions and subtract them from an all-one mask.**
>
> **(2) Specular Reflections**
>
> Following the original 3DGS, we use spherical harmonics (Eq. (2) of our main paper) to model view-dependent appearance variations, including specular reflections. When specular effects are mild, our method remains effective. In rare cases where specular reflections are severe and cause abrupt appearance changes across nearby viewpoints, such regions may be treated as outliers by our masking mechanism. However, this scenario is uncommon when sufficient training views are available. We acknowledge this limitation and plan to explore more robust handling of such extreme cases in future work.
>
>
> [1] Li et al. - 2025 - Segment and Recognize Anything at Any Granularity
>
> [2] Schonberger and Frahm - 2016 - Structure-From-Motion Revisited
>
> ---
> **Q2: How is the self-supervised mask regularized against collapsing to zero?**
>
> The learnable mask is optimized to reflect the feature residual between the rendered and ground-truth images. This residual only becomes zero when the features are orthogonal, which is a rare occurrence. Therefore, the target generally remains non-zero throughout training, making mask collapse unlikely.
>
> In addition, we include a warm-up stage where the 3D representation is sufficiently established. This ensures that in static regions, where the rendered and ground-truth features are highly similar, the residual provides a stable and meaningful supervision signal to guide mask learning.
>
> Lastly, the mask is learned independently for each view. If a mask temporarily collapses in certain regions for one view, signals from other views can still contribute effectively, since multiple views are jointly used in the 3DGS optimization process.
>
> While we did not observe mask collapse in our experiments, we believe including a regularization term is beneficial. We will try an explicit regularization loss that penalizes excessively small mask values (e.g., L2 loss targeting a moderate value), encouraging the mask to remain active and informative throughout training.

---

> ### Author Response · Authors · 2025-08-05
>
> Dear Reviewer REjw,
>
> We sincerely appreciate your detailed review and helpful suggestions. As the discussion period comes to an end, we want to confirm whether our rebuttal and follow-up responses have resolved your concerns. If there are any outstanding issues that still need clarification, we’d be glad to elaborate further.
>
> Thank you.

---

### Official Review · Reviewer_haRb · 2025-07-06

**Clarity:** 3
**Significance:** 3
**Originality:** 3
**Rating:** 4
**Confidence:** 4

**Summary:**

This paper addresses the challenging problem of 3D scene reconstruction from "in-the-wild" image collections, which are often plagued by inconsistent lighting and transient objects (e.g., pedestrians, vehicles) that corrupt the training signal and lead to visual artifacts in the final rendering. Existing methods often rely on heuristic strategies such as per-image appearance embeddings or hand-crafted outlier filtering rules, which can lack stability and generalizability.
Authors propose Asymmetric Dual 3DGS -- a novel framework that enforces consistency between two concurrently trained 3DGS models.
The authors claim state-of-the-art performance in reconstruction quality (PSNR, SSIM, LPIPS) and training efficiency, validated through extensive experiments on the challenging NeRF On-the-go, RobustNeRF, and PhotoTourism datasets. The authors claim state-of-the-art performance in reconstruction quality (PSNR, SSIM, LPIPS) and training efficiency, validated through extensive experiments on the challenging NeRF On-the-go, RobustNeRF, and PhotoTourism datasets.

**Questions:**

* Improve experimental clarity: clarify all ambiguous methodological and experimental details, including the full computational cost of the pipeline (with mask generation), the precise definition of the alternating masking strategy, and provide statistical significance for the main results (e.g., mean and standard deviation over multiple runs)
* It would be great to include sensitivity analysis for the key hyper-parameters
* Strengthen Related Work -- a more nuanced discussion of the related literature, explicitly contrasting the proposed conceptual framework with that of HybridGS would be quite beneficial.

**Ethical Concerns:**

["NO or VERY MINOR ethics concerns only"]

**Limitations:**

yes

**Quality:**

3

**Strengths And Weaknesses:**

Strengths:
* Principled novel core Idea. By leveraging consistency between two independent training processes as a supervisory signal, the method learns to distinguish reliable scene structure from spurious, inconsistent information, which is typically present in the in-the-wild data.
* The asymmetric dual 3DGS framework, with its divergent masking strategy (M_h and Ms), is a particularly smart mechanism for mitigating the risk of "Confirmation Bias" among the different 3DGS models.
* Practicality and technical innovation of the EMA-GS variant -- authors successfully tackle the increased computational budget of the dual 3DGS framework.
* Rigorous & comprehensive empirical validation -- authors compare with various prior methods, including the most recent ones. Table 4 is particularly insightful and includes many important ablations.

Weaknesses:
* The difference and comparison between HybridGS (another dual method) is not articulated well enough.
* Lack of hyper-parameter sensitivity analysis. Specifically, the method's performance hinges on the careful balancing of several loss terms and update rules, controlled by (\lambda_m and \lamba_{mask}). For a method that claims "robustness," it is critical to demonstrate that it is not overly sensitive to its own internal settings.
* Computational cost of mask generation is unclear (was it included into the run-times?)
* The limitations section (in the Supplementary) is quite brief.

---

> ### Author Rebuttal · Authors · 2025-07-30
>
> Thank you for the detailed and constructive review. We greatly appreciate your recognition of the core ideas, practical improvements, and empirical rigor of our work, as well as your thoughtful suggestions for improving clarity, analysis, and comparisons. Your concerns are addressed below.
>
> ---
> **W1: The difference and comparison between HybridGS (another dual method) is not articulated well enough.**
>
> Our approach differs fundamentally from HybridGS in both design and training. HybridGS separates static and dynamic content using two models (3DGS for static, 2DGS for dynamic) and requires a staged training process with a learnable blending mask. In contrast, our method uses a dual 3DGS setup with mutual supervision to improve robustness against dynamic noise, all within the standard 3DGS training pipeline. While both methods use masking, HybridGS blends outputs based on 2DGS-derived uncertainty, whereas we apply two distinct masking strategies to reduce confirmation bias from a single, potentially inaccurate mask.
>
> We will elaborate more on these differences in the main paper.
>
> ---
> **W2: Lack of hyper-parameter sensitivity analysis. Specifically, the method's performance hinges on the careful balancing of several loss terms and update rules, controlled by ($\lambda_m$ and $\lambda_{mask}$). For a method that claims "robustness," it is critical to demonstrate that it is not overly sensitive to its own internal settings.**
>
> **Table A: Performance under varying weights of the mutual consistency loss.**
>
> | Dataset | On-the-go  |  |  | RobustNeRF |  |  |
> |---|---|---|---|---|---|---|
> | $\lambda_m$  | PSNR | SSIM | LPIPS | PSNR | SSIM | LPIPS |
> | 0 | 23.10 | 0.801 | 0.132 | 29.72 | 0.915 | 0.081 |
> | 0.05 | 23.39 | 0.803 | 0.136 | 30.20 | 0.917 | 0.080 |
> | 0.1 | 23.40 | 0.801 | 0.135 | 30.27 | 0.917 | 0.080 |
> | 0.2 | 23.43 | 0.805 | 0.133 | 30.30 | 0.917 | 0.081 |
> | 0.3 | 23.47 | 0.805 | 0.134 | 30.24 | 0.917 | 0.081 |
>
>
> **Table B: Performance under varying weights of the learnable mask loss.**
> | Dataset | On-the-go  |  |  | RobustNeRF |  |  |
> |---|---|---|---|---|---|---|
> | $\lambda_{mask}$ | PSNR | SSIM | LPIPS | PSNR | SSIM | LPIPS |
> | 0.5 | 23.33 | 0.802 | 0.134 | 30.20 | 0.916 | 0.080 |
> | 1.0 | 23.40 | 0.801 | 0.135 | 30.27 | 0.917 | 0.080 |
> | 1.5 | 23.39 | 0.804 | 0.135 | 30.20 | 0.917 | 0.080 |
> | 2.0 | 23.32 | 0.802 | 0.134 | 30.21 | 0.916 | 0.081 |
>
> As shown in Table A and B, although the best performance is generally achieved at our default setting ($\lambda_m = 0.1$ and $\lambda_{mask} = 1.0$), the differences across settings are minimal (less than 0.1 dB). This indicates that the performance is not highly sensitive to the values of $\lambda_m$ and $\lambda_{mask}$. When $\lambda_m = 0$, mutual consistency regularization is disabled, resulting in a significant performance drop. Due to time constraints, all experiments above were conducted using the EMA-GS setting, and the PhotoTourism dataset was excluded.
>
> ---
> **W3: Computational cost of mask generation is unclear (was it included into the run-times?)**
>
> **Table C: Average per-scene running time across different datasets (in hours).**
>
> | Time\Dataset | On-the-go  | RobustNeRF  | PhotoTourism |
> |---|---|---|---|
> | Mask Generation  | 0.06 | 0.07 | 0.56 |
> | Training (GS-GS)  | 0.28 | 0.31 | 5.30 |
> | Training (EMA-GS)  | 0.18 | 0.21 | 2.90 |
>
> No, it is not included in the overall runtime. Mask generation is a one-time preprocessing step performed offline, similar to COLMAP in existing 3DGS pipelines, which is also excluded from training time. We present the mask generation times in Table C. Notably, the PhotoTourism dataset requires significantly more time due to its higher image resolution and greater number of views.
>
> ---
> **W4: The limitations section (in the Supplementary) is quite brief.**
>
> We follow WildGaussians in using a per-view appearance embedding to control global appearance, and a per-Gaussian embedding shared across views to model the appearance of individual Gaussian primitives. While this strategy is effective for handling dynamic lighting variations across training frames, it may struggle to capture fine-grained effects such as object highlights. This can happen when the per-view and intrinsic appearances are not well decoupled, where the per-view embedding incorrectly attributes static intrinsic appearance to dynamic lighting, and the per-Gaussian embedding lacks the capacity to model high-frequency details. While our proposed mutual consistency loss helps improve this decoupling, it does not completely resolve the issue. This limitation is common in many existing works [1,2,3] and remains an important direction for future research.
>
> [1] Fridovich-Keil et al. - 2023 - K-Planes: Explicit Radiance Fields in Space, Time, and Appearance
>
> [2] Dahmani et al. - 2025 - SWAG: Splatting in the Wild Images with Appearance-Conditioned Gaussians
>
> [3] Kulhanek et al. - 2024 - WildGaussians: 3D Gaussian Splatting In the Wild
>
> ---
> **Q1: Improve experimental clarity: clarify all ambiguous methodological and experimental details, including the full computational cost of the pipeline (with mask generation), the precise definition of the alternating masking strategy, and provide statistical significance for the main results (e.g., mean and standard deviation over multiple runs).**
>
> **(1) The full computational cost of the pipeline is discussed in the response to Weakness 3.**
>
> **(2) Clarification on the alternating masking strategy**
>
> In the GS-GS setting, we apply distinct masks to each branch of the dual 3DGS models as shown in Eq. (8). However, this is not feasible in the EMA-GS setting, which uses only a single 3DGS model and maintains an exponential moving average for consistency regularization. To encourage diverse supervision, we adopt an alternating masking strategy described in Eq. (11): one mask is used during a given optimization step, and the other is applied in the following step, alternating throughout training. This approach mitigates the bias of relying on a single mask.
>
> **(3) Statistical significance for the main results**
>
> **Table D: Quantitative results on the NeRF On-the-go dataset. Each experiment is repeated five times, and we report the mean and standard deviation.**
>
> | Setting |GS-GS | | | EMA-GS | | |
> |------------------|--------|--------|--------|--------|--------|--------|
> | Scene | PSNR | SSIM | LPIPS | PSNR | SSIM | LPIPS |
> | High Occlusion | 24.36 ± 0.02 | 0.823 ± 0.001 | 0.151 ± 0.001 | 24.11 ± 0.05 | 0.819 ± 0.002 | 0.152 ± 0.004 |
> | Medium Occlusion | 24.52 ± 0.06 | 0.871 ± 0.001 | 0.090 ± 0.001 | 24.26 ± 0.08 | 0.864 ± 0.001 | 0.092 ± 0.002 |
> | Low Occlusion | 21.99 ± 0.04 | 0.730 ± 0.001 | 0.184 ± 0.004 | 21.81 ± 0.09 | 0.723 ± 0.002 | 0.166 ± 0.007 |
>
> **Table E: Quantitative results on the RobustNeRF dataset. Each experiment is repeated five times, and we report the mean and standard deviation.**
> | Setting |GS-GS | | | EMA-GS | | |
> |------------------|--------|--------|--------|--------|--------|--------|
> | Scene | PSNR | SSIM | LPIPS | PSNR | SSIM | LPIPS |
> | Statue | 23.44 ± 0.05 | 0.893 ± 0.001 | 0.098 ± 0.001 | 23.46 ± 0.06 | 0.890 ± 0.001 | 0.097 ± 0.002 |
> | Android| 25.58 ± 0.05 | 0.856 ± 0.001 | 0.070 ± 0.003 | 25.47 ± 0.06 | 0.849 ± 0.002 | 0.070 ± 0.002 |
> | Yoda | 37.12 ± 0.09 | 0.969 ± 0.001 | 0.074 ± 0.001 | 36.46 ± 0.06 | 0.967 ± 0.001 | 0.078 ± 0.001 |
> | Crab | 36.11 ± 0.07 | 0.963 ± 0.001 | 0.079 ± 0.001 | 35.52 ± 0.07 | 0.961 ± 0.001 | 0.080 ± 0.001 |
>
> We repeated the experiment for five time. We did not include results on the PhotoTourism dataset due to time constraints, but we will include it in the final version. Based on the results in Table D and E, our method shows statistically significant improvements.
>
> ---
> **Q2: It would be great to include sensitivity analysis for the key hyperparameters.**
>
> Please refer to Weakness 2, we will revise it accordingly.
>
> ---
> **Q3: Strengthen Related Work -- a more nuanced discussion of the related literature, explicitly contrasting the proposed conceptual framework with that of HybridGS would be quite beneficial.**
>
> Please refer to Weakness 1, we will revise it accordingly.

---

> ### Author Response · Authors · 2025-08-05
>
> Dear Reviewer haRb,
>
> Thank you for taking the time to review our work and engage in the discussion. As we approach the end of the discussion phase, we would like to ensure that all of your concerns have been adequately addressed. If anything remains unclear or requires further explanation, please don’t hesitate to let us know.
>
> Thank you.

---

### Note · Authors · 2025-08-12

We sincerely thank all reviewers for their thoughtful feedback and recognition of our contributions. We are encouraged by the affirmation of the following contributions:
1. The novelty of using two asymmetric 3DGS models to improve robustness under in-the-wild conditions (haRb, Rejw, b1d5, 8u6B)
2. The use of two distinct masking strategies to mitigate confirmation bias (haRb, Rejw, b1d5, 8u6B)
3. The practicality and technical innovation of the EMA-GS variant (haRb, Rejw, b1d5, 8u6B)
4. The comprehensiveness and clarity of our ablation studies (haRb, Rejw, b1d5, 8u6B)

We also appreciate the constructive suggestions and have addressed all concerns in our rebuttal:
1. Clarified the distinctions between our method and HybridGS (haRb, Rejw, b1d5, 8u6B)
2. Added experimental evidence demonstrating the statistical significance of our results (haRb)
3. Provided additional details on the proposed masking strategies (haRb, Rejw, b1d5, 8u6B), appearance modeling (Rejw), and the stochastic nature of artifacts (8u6B)

Thank you again for your time, effort, and valuable insights.

---

### Decision · Program_Chairs · 2025-09-17

**Decision:**

Accept (spotlight)

**Comment:**

Based on the discussion on the stochastic nature of training 3DGS by its randomness, they proposed an asymmetric dual 3DGS. Divergent masking techniques, i.e., multi-cue adaptive and self-supervised soft masks, are applied to ensure the asymmetrical models. One clever trick is one of two asymmetric models is replaced with EMA Gaussians to avoid directly training two models and a masking strategy is alternatively applied to give stochasticity. AC thought this interesting approach may need some attention from the community for further developments. Strong experimental validation and ablation studies support this idea.

There are no remaining major issues, and two reviewers recommend a “spotlight” presentation for its novelty. AC also enjoyed reading this paper and glad to recommend it as a spotlight.

AC strongly encourages you to revise the manuscript in accordance with the rebuttal responses and to responsibly prepare the release of the code for the benefit of the community.

Minor comments:
- You should correctly refer to the corresponding section of Appendix, especially, for Multi-Cue Adaptive Mask in the manuscript. Recommend that every sections, tables, abd figures are appropriately refered in the manuscript. This will greatly help for readers.
- Section titles should be in lower cases except for the first word and other special cases according to the official NeurIPS style guideline. “All headings should be lower case (except for first word and proper nouns)” in Line 55. Major ML/vision conferences adopt this style except some.